# Membrane cholesterol mediates the cellular effects of monolayer graphene substrates

Kristina E. Kitko[1,2], Tu Hong[3], Roman M. Lazarenko [1], Da Ying[3], Ya-Qiong Xu[2,3,4] & Qi Zhang[1,2]

Graphene possesses extraordinary properties that promise great potential in biomedicine. However, fully leveraging these properties requires close contact with the cell surface, raising the concern of unexpected biological consequences. Computational models have demonstrated that graphene preferentially interacts with cholesterol, a multifunctional lipid unique to eukaryotic membranes. Here we demonstrate an interaction between graphene and cholesterol. We find that graphene increases cell membrane cholesterol and potentiates neurotransmission, which is mediated by increases in the number, release probability, and recycling rate of synaptic vesicles. In fibroblasts grown on graphene, we also find an increase in cholesterol, which promotes the activation of P2Y receptors, a family of receptor regulated by cholesterol. In both cases, direct manipulation of cholesterol levels elucidates that a graphene-induced cholesterol increase underlies the observed potentiation of each cell signaling pathway. These findings identify cholesterol as a mediator of graphene's cellular effects, providing insight into the biological impact of graphene.

[1] Department of Pharmacology, Vanderbilt University, Nashville, TN 37232, USA. [2] Program in Interdisciplinary Materials Science, Vanderbilt University, Nashville, TN 37235, USA. [3] Department of Electrical Engineering and Computer Science, Vanderbilt University, Nashville, TN 37235, USA. [4] Department of Physics and Astronomy, Vanderbilt University, Nashville, TN 37235, USA. Kristina E. Kitko, Tu Hong and Roman M. Lazarenko contributed equally to this work. Correspondence and requests for materials should be addressed to Q.Z. (email: qi.zhang@vanderbilt.edu)

Graphene is a two-dimensional material composed of a single-layer of hexagonal $sp^2$-hybridized carbon atoms[1]. A consequence of its unique atomic structure, graphene possesses a myriad of attractive chemical and physical properties: exceptionally high electron mobility, thermal conductivity, optical transmittance, mechanical strength, chemical stability, and surface area-to-volume ratio[2,3]. This combination of features has made graphene a promising material for a broad range of biomedical applications, including drug delivery, tissue engineering, biosensing, and neuroprosthetics[4–6]. Furthermore, its superior carrier mobility enables graphene-based electrodes to detect electrochemical changes associated with a variety of cellular activities or to deliver optical or electrical stimuli. For example, graphene electrodes have been used as voltage sensors to measure membrane potential changes at the single-cell level and to record electrical activity in neuronal networks in vitro and in vivo[7,8]. However, a significant drawback to the use of graphene-based devices remains, in that detection efficiency exponentially decreases as the distance from the cells or tissue increases. Therefore, graphene needs to be close to cell or tissue surfaces[8] in order to maximize its utility in bioapplications. This then raises a fundamental question: how graphene affects the plasma membrane and related cellular functions.

Prior studies have documented that graphene flakes (GFs) are destructive to Gram-negative bacteria such as *Escherichia coli* through the disruption of plasma membrane integrity[9–11]. Computational modeling and electron microscopy have suggested that this is the result of nanoscale GFs penetrating the bacterial plasma membrane and dispersing phospholipids[12], which leads to membrane disintegration and cell death[13]. Interestingly, no such cytotoxicity has been reported in eukaryotic cells. Instead, cell proliferation[14], differentiation[15], and morphogenesis[16] were improved when graphene with traditional tissue culture coatings was used as a culture substrate. For mouse neurons grown on bare graphene, Veliev et al.[17] demonstrated that the crystallinity of graphene promotes cell adhesion and neurite outgrowth, suggesting the significance of close contact between graphene and the cell surface in mediating graphene's effects. Aggregated GFs used as a substrate or applied acutely to mature neuronal cultures resulted in few developmental or morphological changes[18–20]. Aggregated GFs, however, have very different surface characteristics and physicochemical properties from monolayer graphene substrates[21]. Given the variation in experimental outcomes and the increasing interest in bioapplications of graphene, it remains important to understand how graphene interacts with the eukaryotic plasma membrane and how its diverse cellular effects are realized.

We reasoned that, since the plasma membrane is the initial point via which any downstream cellular responses to graphene would be realized, membrane-associated molecule(s) were responsible for the variation of cellular effects in response to graphene. We hypothesized that cholesterol was involved for two reasons: it is unique to and abundant in the eukaryotic plasma membrane but absent from prokaryotic membranes, and it is related to many cellular processes, realized through both direct and indirect interactions with other lipids and proteins[22]. For example, cholesterol influences membrane fluidity[23], facilitates exocytosis/endocytosis[24], regulates integral membrane proteins like G protein-coupled receptors (GPCRs)[25], and organizes cytoskeletal attachment as well as cell adhesion[26]. The distribution of cholesterol within the cell membrane is highly heterogeneous and dynamic[22], with trafficking between surface and intracellular membranes helping to maintain overall cholesterol homeostasis. Intriguingly, it has been empirically demonstrated that the planar tetracyclic ring structure of cholesterol may permit it to stack on the graphene surface[27–31]. We thus sought to investigate if and how the cellular effects of monolayer graphene are related to membrane cholesterol.

Here we use liquid-phase exfoliation (LPE) of graphene powder to produce suspended GFs, which enables solution-based measurements to investigate a graphene–cholesterol interaction. GFs extract cholesterol from cholesterol-containing culture media and effectively quench a fluorescent cholesterol analog, suggesting an interaction between graphene and cholesterol. For cell-based studies, we grew both neurons and fibroblast cells on large-area monolayer graphene sheets. In neurons on graphene, we find increased cell membrane cholesterol and significant increases in the number, the release probability, and the turnover rate of synaptic vesicles—all of which are modulated by membrane cholesterol[32–35]. Bidirectional manipulations of membrane cholesterol demonstrate that a graphene-induced membrane cholesterol increase is an underlying mechanism for the potentiated neurotransmitter release. Furthermore, we demonstrate that graphene substrates allosterically enhance the $Ca^{2+}$ responses of P2Y receptors to ATP stimulation in a cholesterol-dependent manner. Our findings collectively reveal that cholesterol is a key mediator of graphene's biological effects on eukaryotic cells.

## Results

**Characterizing an interaction between graphene and cholesterol.** To test graphene's ability to attract cholesterol, we produced GFs from graphite powder using a well-established LPE protocol[36]. GFs were suspended in 2 wt% polyvinylpyrrolidone (PVP) at 260 ng mL$^{-1}$ to prevent aggregation. We mixed this suspension with serum-containing neuronal culture media and incubated for 24 h to mimic chronic exposure during culture. Serum in our culture media contains cholesterol (see result below and Methods), which enabled us to measure cholesterol adsorption onto the graphene surface. GFs were separated out by size-dependent filtration and subjected to an enzymatic cholesterol assay[37]. The remaining fraction, untreated fresh media, and media mixed with PVP (as a vehicle control) were assayed in the same manner. After calibration by a cholesterol standard, we determined that cholesterol concentrations in the PVP-treated media (12.4 ± 2.3 μM) and untreated fresh media (12.3 ± 0.9 μM) were similar. The GF fractions (graphene incubated with media) showed a significantly higher cholesterol concentration (18.0 ± 1.0 μM), while there was a correspondingly lower concentration (4.3 ± 0.1 μM) in the remaining fraction (graphene-treated media) (Fig. 1a), suggesting that GFs extract cholesterol from culture media. To verify that this was due to an interaction between graphene and cholesterol rather than the non-specific adsorption of proteins that cholesterol is complexed with, we used a fluorescent cholesterol analog, TopFluor Cholesterol (TFC, a.k.a. BODIPY-cholesterol). TFC is biophysically similar to cholesterol and thus can be used to study cell membrane cholesterol in live cells[38–40]. Since graphene is a highly efficient acceptor in Förster resonance energy transfer[41], we reasoned that TFC fluorescence would be quenched if an interaction stabilized cholesterol on the GFs surface. In line with this prediction, spectrofluorometric measurements demonstrated that GFs significantly reduced TFC fluorescence in a concentration- and time-dependent manner (Fig. 1b and Supplementary Figure 1). Furthermore, GFs had little effect on BODIPY fluorescence (Fig. 1b), suggesting the interaction between GFs and TFC is specific to the cholesterol group.

**Production and characterization of graphene substrates.** Having demonstrated an interaction between GFs and cholesterol, we set out to test whether this interaction plays a role in mediating graphene's cellular effects. We produced large films of planar graphene via chemical vapor deposition (CVD), which were

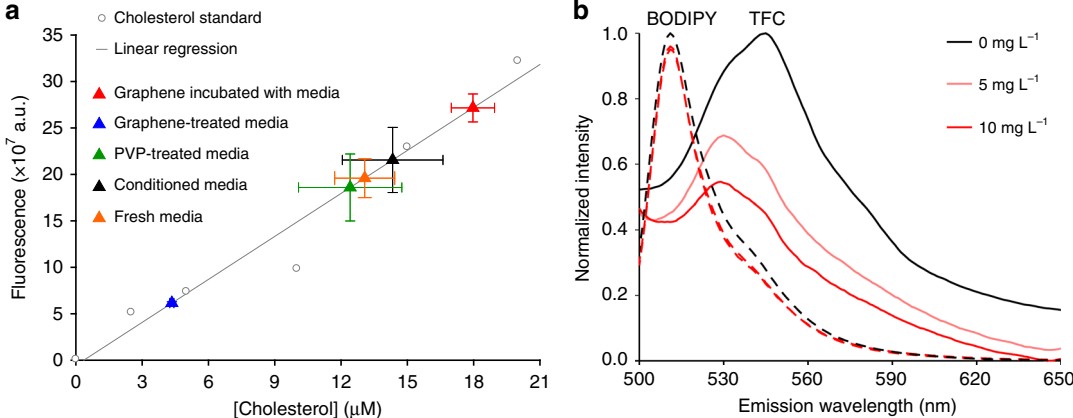

**Fig. 1** Graphene interacts with cholesterol. **a** Cholesterol concentrations (based on the working curve generated from cholesterol standards, gray circles and line) in fresh serum-containing (orange triangle), untreated (black triangle), PVP-treated (green triangle), graphene-treated media (blue triangle), and graphene incubated with media (red triangle). $n = 4$ samples per batch, $N = 4$ total batches. Error bars are S.E.M. **b** Spectrofluorometer measurements of BODIPY (dashed line) and TFC (solid line) emission after 1-h incubation with graphene (light or dark red), adjusted for the concentration-dependent broadband absorbance of graphene (Supplementary Figure 1) and normalized to the maximum value of 0 mg L$^{-1}$ graphene and minimum value of 10 mg L$^{-1}$ graphene at each measured wavelength for both dyes

solution transferred to bare glass coverslips to be used as culture substrates (Fig. 2a). The quality of the CVD graphene was examined by Raman spectroscopy. The characteristic 2D and G peaks as well as the large ratio between them[42,43] suggest that the graphene films are monolayer (Fig. 2b). Spatially resolved Raman intensity maps of the G and 2D peaks further demonstrate their uniformity (Fig. 2c, d). Because previous reports have demonstrated that neurons remain viable on uncoated graphene during long-term culture[17–19], we omitted Matrigel, an extracellular substrate commonly used for hippocampal cultures (see Methods), and plated dissociated neurons directly on pristine graphene. Conventional hippocampal cultures contain a considerable number of astrocytes[44], which if allowed to proliferate over the length of the culture period would have covered the substrate surfaces, further limiting the exposure of neurons to graphene. To minimize potential effects from the astrocyte layer, we applied a mitotic inhibitor (Ara-C) immediately after cell plating, which blocks astrocyte proliferation without any effect on non-proliferative cells (i.e., neurons)[44]. However, astrocytes play important developmental and functional roles for neurons, so to compensate for the reduced number of astrocytes, we routinely supplied cultures with conditioned media (see Methods) harvested from conventional hippocampal cultures prepared in parallel. As a negative control, the same astrocyte-deprived hippocampal cultures were grown on bare glass coverslips and maintained in the same manner.

It is well established that nanomaterial surfaces will be covered by a variety of biomolecules (i.e., a protein corona) after introduction to any biological system. Although we omitted a much thicker artificial protein-coating layer (Matrigel) in plating our cultures, we cannot exclude the likelihood of the formation of a protein corona on the graphene surface. However, our cell-free assays demonstrate that cholesterol enrichment on graphene still occurs even in the likely presence of a protein corona after 24-h incubation (Fig. 1a). We performed Raman spectroscopy on neurons grown on graphene and found that the intensity of both 2D and G peaks were suppressed in cell-containing regions (Fig. 2h–j), similar to what was observed in Matrigel-coated areas (Fig. 2e–g), suggesting that plated neurons change graphene's Raman spectra in the same manner that Matrigel does.

**Increased cholesterol in neurons on graphene substrates**. Computational studies predict that graphene oriented parallel to

the cell surface can induce a local enrichment of cholesterol[29,30]. To experimentally investigate possible changes in cholesterol after chronic growth on graphene, we first performed Filipin staining, a conventional method for labeling cholesterol in fixed cells[45]. We found a 27% increase in Filipin staining intensity in the neurites of neurons cultured on graphene (Fig. 3a, b), suggesting that graphene indeed increased cholesterol in neurites. To determine whether a cholesterol increase occurred in live neurons, we turned to generalized polarization (GP) imaging, which uses a ratiometric reporter based on a fluorescent dye sensitive to membrane fluidity. Since membrane fluidity is in turn inversely correlated to membrane cholesterol concentration, GP imaging is commonly used to indirectly measure cholesterol changes in the cell membrane[46]. We used C-laurdan, a more sensitive and photostable derivative of Laurdan[47]. GP value was calculated as $(I_{blue} - G \times I_{green})/(I_{blue} + G \times I_{green})$, in which $G$ is a correction factor and $I_{blue}$ and $I_{green}$ are fluorescence intensity values[46]. We found that C-laurdan fluorescence exhibited a blue shift on graphene, which translated to an overall shift toward increased GP values, indicating increased plasma membrane cholesterol on graphene substrates (Fig. 3c, d).

Given that TFC has been demonstrated to behave similarly to endogenous cholesterol in membrane incorporation and phase partitioning[38–40], we used TFC to mimic the membrane distribution of endogenous cholesterol and thus to examine graphene's effect on cell membrane cholesterol. Because of reported differences between TFC and cholesterol in intracellular transportation and lysosomal accumulation[39,40], we used a shorter loading time (1 h) and lower concentration (1 µM) of TFC; this should largely result in TFC incorporation into the plasma membrane. However, we cannot exclude the possibility that membrane turnover will also result in some intracellular membrane labeling. We found that TFC is distributed throughout neurites (Fig. 4a). Intriguingly, neurons on graphene showed significantly increased TFC labeling (Fig. 4b). As synaptic boutons generally have a higher cholesterol concentration than other parts of the neuronal membrane[48,49], which might thus limit the ability of the membrane to incorporate cholesterol analog, we then asked whether graphene could also increase TFC's membrane insertion within synaptic boutons. To this end, we measured TFC intensity in areas defined by FM4-64 labeling (i.e., synaptic boutons) (Fig. 4c, d, arrow heads), a far-red fluorescent dye that preferentially labels synaptic vesicles[50]. Since

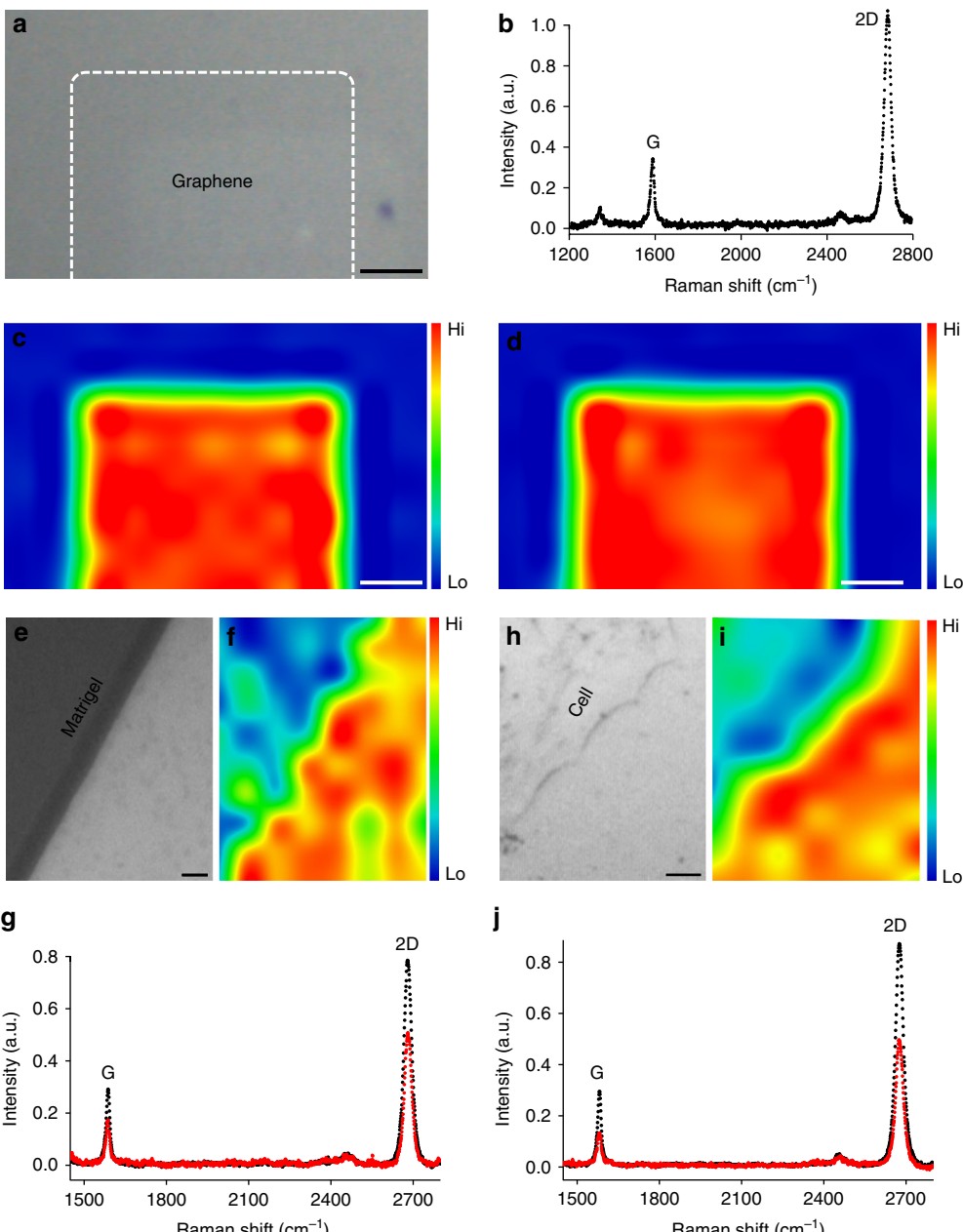

**Fig. 2** Characterization of graphene films. **a** Bright field image shows the field of view. Scale bar, 1 mm. **b** Raman spectrum at the graphene-covered area shows characteristic G and 2D peaks. The high 2D vs. G ratio and a symmetric 2D peak are consistent with those of monolayer graphene. Corresponding spatially resolved map of **c** Raman G-peak intensity and **d** 2D-peak intensity from the same field of view. Scale bar, 1 mm. **e** Bright field image shows the edge of a Matrigel droplet (upper-left) on a graphene-coated glass coverslip. **f** Corresponding Raman 2D-peak intensity map. Scale bar, 5 μm. **g** Raman spectrum of the bare graphene area (black) shows a high 2D vs. G ratio and a symmetric 2D peak, indicating monolayer graphene. At the Matrigel-coated graphene area (red), there is a strong reduction of intensity of both the G and 2D peaks. **h** Bright field image shows the edge of a cell (upper left) growing on a graphene-coated glass coverslip. **i** Corresponding Raman 2D-peak intensity map. Scale bar, 5 μm. **j** Raman spectrum at the cell-covered graphene area (red) also exhibits an intensity reduction for both G and 2D peaks in comparison with that at the bare graphene area (black)

an increase of synaptic vesicles can increase the total presynaptic membrane area and consequently TFC staining, we examined the relationship between TFC staining and FM4-64 staining (Fig. 4e, scatter plots and regression fits). The overall increase of TFC intensity relative to increases in FM4-64 intensity was higher on graphene (Fig. 4e), suggesting that graphene increases membrane cholesterol regardless of possible changes in synaptic vesicle numbers at synaptic boutons.

**Physiological properties of neurons on graphene**. To probe whether an increase in cholesterol elicited functional effects, we recorded synaptically connected neurons on graphene or glass between 13 and 17 DIV (days in vitro) (Fig. 5a). The amplitude of spontaneous excitatory postsynaptic currents (sEPSCs) was slightly but not significantly larger in neurons on graphene (Fig. 5b), while sEPSC frequency was significantly higher (i.e., a shorter inter-event interval) (Fig. 5c). Glutamate is the major excitatory neurotransmitter in our culture configuration and sEPSCs are largely mediated by two ionotropic glutamate receptors: α-amino-3-hydroxy-5-methyl-4-isoxazolepropionic acid receptors (AMPARs) and N-methyl-D-aspartate receptors (NMDARs), which have distinct activation and decay kinetics

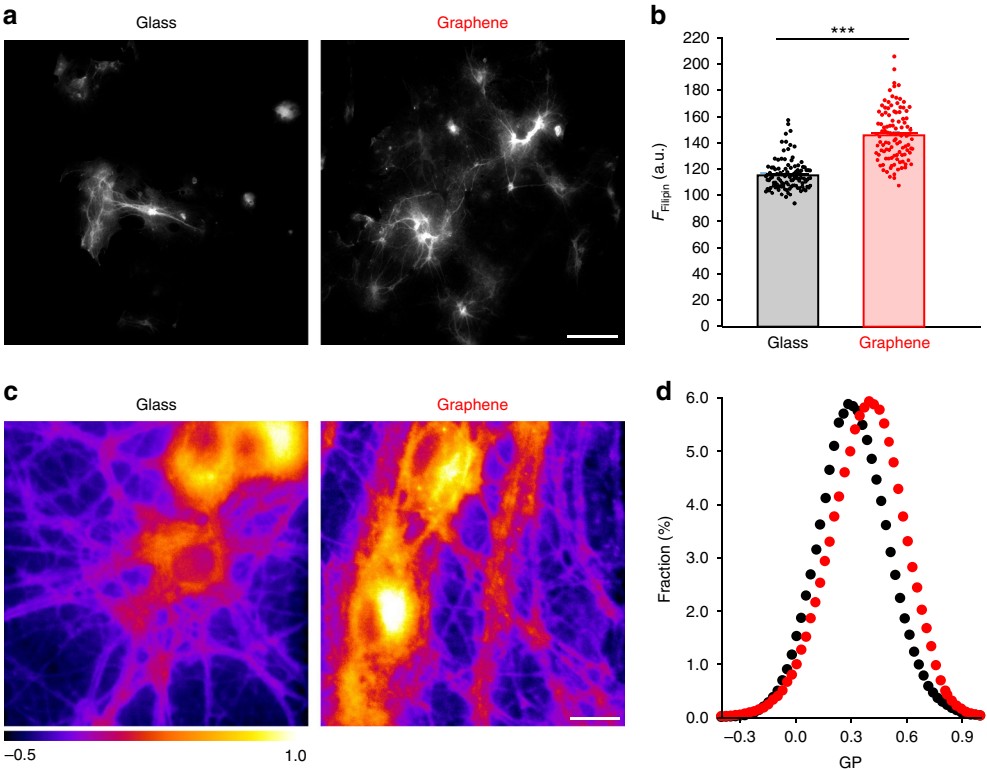

**Fig. 3** Graphene increases cell membrane cholesterol. **a** Sample images of Filipin staining. Scale bar, 100 μm. **b** Filipin fluorescence intensity in neuronal neurites ($n_{glass} = 118$ and $n_{graphene} = 112$ neurites, $N = 3$ batches, ***$p < 0.001$, Wilcoxin rank-sum test, Cohen's $d = 1.89$). **c** Sample generalized polarization (GP) images. Scale bar, 20 μm. **d** Distributions of GP values over individual image pixels ($n = 7$ FOVs, $N = 3$ batches; $p < 0.05$, Kolmogorov–Smirnov test of the distributions of GP values, see 'Statistical analysis' section in Methods). Error bars are S.E.M.

(Supplementary Figure 2). Because changes in sEPSC frequency could be modulated through changes in presynaptic neurotransmitter release or by changes in the postsynaptic composition of NMDARs and AMPARs (measured as the $I_{NMDAR}$ vs. $I_{AMPAR}$ ratio[51]), to determine whether the increased sEPSC frequency on graphene was driven by postsynaptic mechanisms, we measured $I_{NMDAR}/I_{AMPAR}$. Here we observed a small but non-significant decrease in this ratio (Fig. 5d). Taken together with a small but non-significant increase in sEPSC amplitude, this suggested that the effects of graphene are presynaptic.

**Morphological characterization of neurons on graphene**. To understand the cellular basis of our observed electrophysiological changes, we performed immunofluorescence labeling using antibodies against a selective synaptic vesicle marker, Synaptophysin (Syp), and a neurite-specific marker, β-III Tubulin (Tuj1). Recorded neurons were retrospectively identified by biocytin infused through the patch electrodes (Fig. 6a). We first examined neurite complexity, as complexity is correlated to synaptic connectivity[52]. It is generally established that increased neurite complexity is associated with more synaptic connections between neurons, and more synaptic connections in turn lead to more synaptic inputs (i.e., higher sEPSC frequency)[53]. We performed Sholl analysis to quantify arborization, which counts the number of neurite intersections at concentric circles of increasing radius originating at the neuronal soma[54]. Although there was an overall increase in neurite complexity on graphene between 12 and 18 DIV (Fig. 6c and Supplementary Table 1), the effect was small (see Fig. 6c legend) and furthermore not significant at any distance. However, we did note a non-significant increase on graphene at 68 μm. To determine whether the overall increase was a

result of early developmental changes, we performed Sholl analysis on neurons at 3 or 7 DIV. Here we found no differences in overall complexity (i.e., number of intersections); there were also only very small increases in Syp staining intensity (Supplementary Figure 3). Collectively, our analysis of neurite complexity throughout development suggests that graphene does not substantially alter early outgrowth (up to 7 DIV) but does slightly increase the overall dendritic arbor during late phase development (7–12 DIV). To determine whether neurons on graphene had more synaptic connections per neurite, we measured the lateral density of synaptic boutons along neurites (the number of Syp-positive puncta in a unitary length of Tuj1- and biocytin-positive process) (Fig. 6b). The lateral density of boutons was not altered on graphene (Fig. 6d), in line with the idea that developmental changes on graphene are limited and do not act to alter synaptic connectivity in mature neurons.

Given that our functional data documented synaptic potentiation on graphene (Fig. 5c), we next examined Syp immunostaining. Syp is highly localized to within synaptic vesicles[55], thus differences in Syp staining intensity are positively correlated to differences in the number of vesicles within presynaptic terminals. We found that the mean fluorescence of Syp-positive puncta was ~21% greater in the graphene group (Fig. 6e). We also analyzed the standard deviation of the pixel intensity (i.e., clustering index) within individual Syp-positive puncta. Within an individual region of interest (ROI; see Methods), distributed vesicles would lead to a more uniform overall intensity, thus a lower standard deviation—i.e., a lower clustering index. Conversely, more clustered vesicles would be less uniform, resulting in a larger clustering index within individual boutons[49]. We found a significantly increased average clustering index across all

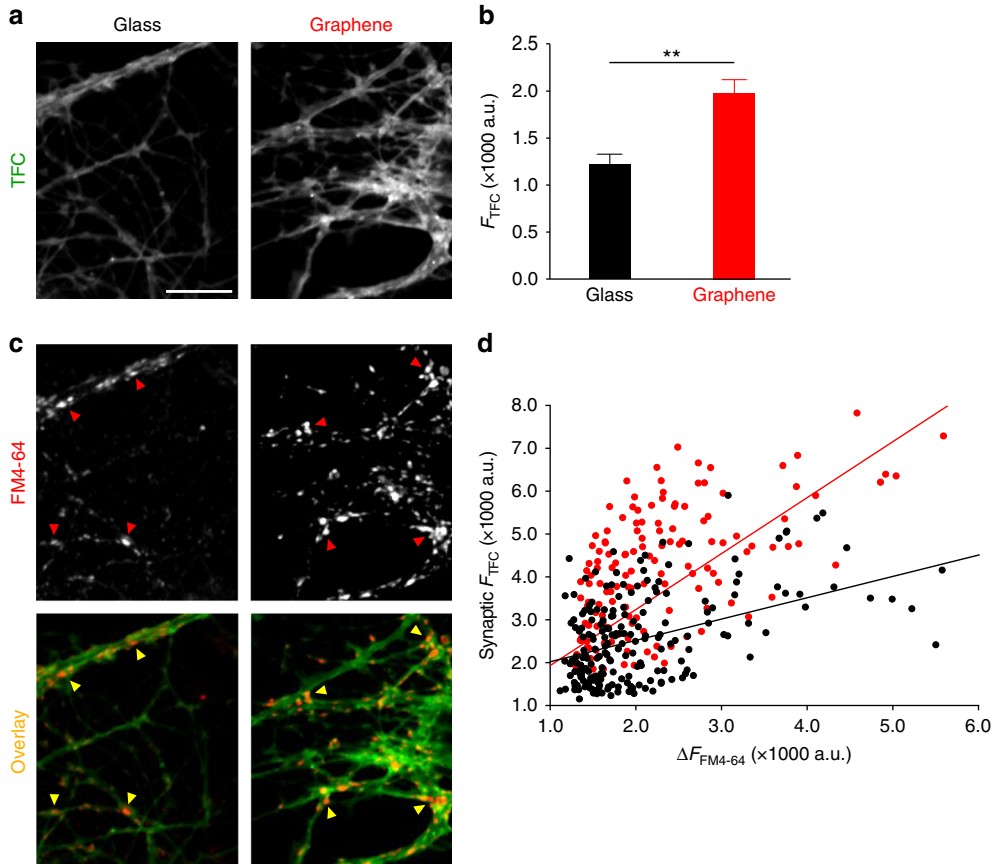

**Fig. 4** Graphene increases synaptic membrane cholesterol. **a** Sample images of TopFluor-Cholesterol (TFC) staining of neurons. Scale bar, 10 µm. **b** Average TFC staining intensity of threshold defined ROIs ($n = 9$ FOVs, $N = 3$ batches, **$p < 0.01$, two-tailed $t$-test). **c** Sample images of FM4-64 staining of neurons. Same scale bar as **a**. Arrowheads indicate examples of synaptic boutons defined by FM4-64 labeling. **d** Overlay of **a** and **b**, arrowheads from **b**. **e** ΔFM 4-64 fluorescence intensity (see Methods) vs. TFC intensity in FM4-64-defined synaptic boutons (both $n = 9$ FOVs, $N = 3$ batches; $p < 0.05$, two-tailed $t$-test). Linear regression fittings indicate correlation for graphene ($F_{TFC} = 1.2083 \times F_{FM4-64} - 253.29$ a.u., Pearson correlation coefficient = 0.6629, red solid line) and glass ($F_{TFC} = 0.6273 \times F_{FM4-64} + 837.85$ a.u., Pearson correlation coefficient = 0.5698, black solid line). Error bars are S.E.M.

ROIs on graphene (Fig. 6f), suggesting that synaptic vesicles are more tightly clustered in individual presynaptic terminals.

**Graphene substrates potentiate neurotransmitter release.** Our electrophysiological and immunolabeling results collectively suggested synaptic vesicle changes in presynaptic terminals. Because cholesterol is essential for synaptic vesicle origination, distribution, and turnover[56], we turned our attention to changes in synaptic vesicles on graphene substrates. To study synaptic vesicles, we used FM1-43, a styryl dye with better kinetic properties than FM4-64[57]. The amount of FM1-43 uptake reflects the number of the releasable synaptic vesicles[57]. In good agreement with FM4-64 loading (Fig. 4e) and Syp staining (Fig. 6e), there was significantly more FM1-43 uptake (~27%) by neurons on graphene (Fig. 7a, b, inset), suggesting an increase in releasable vesicles. Subsequent high K⁺ stimulation with a dye-free external solution causes FM1-43 loss from synaptic vesicles as they undergo evoked exocytosis. The amount and the rate of dye loss reflect synaptic vesicle release probability[57]. We observed a ~30% increase in total dye loss in neurons on graphene (Fig. 7c, d, inset) and an increase in the rate of dye loss (Supplementary Figure 4), indicating an increased pool of releasable vesicles and an increase in their release probability. Together, our results suggest that graphene leads to an increase of releasable vesicles, which acts to potentiate neurotransmission.

To further elucidate the mechanisms underlying this presynaptic potentiation, we performed single vesicle imaging using quantum dots (Qdots), an approach which provides a more precise estimate of releasable synaptic vesicle amounts and their release probability. We began by loading Qdots into all releasable vesicles (the total releasable pool (TRP)) using a combination of a high concentration of Qdots (100 nM) and strong stimulation (2-min 90 mM K⁺)[58,59] (Fig. 7e). Based on the unitary photoluminescence of a single Qdot (see Supplementary Discussion), we estimated that the average numbers of TRP vesicles per synapse were $23.2 \pm 0.4$ on glass and $29.2 \pm 0.6$ on graphene, a ~26% increase (Fig. 7f). We next applied Qdots at a low concentration (0.8 nM) to randomly load single vesicles across the TRP (Fig. 7g). Quantal analysis of Qdot photoluminescence[58,59] in FM4-64-defined synaptic boutons confirmed that a large fraction of terminals were loaded with single Qdots inside individual synaptic vesicles (Supplementary Figure 5a). We stimulated neurons for 1-min with high K⁺ solution and imaged Qdot release from synaptic vesicles. The size and inherent pH sensitivity of Qdots allows for discrimination between recycling modes via patterns of photoluminescence changes (Supplementary Figure 5b): a small and transient increase in Qdot photoluminescence alone indicates fast and reversible fusion (FRF), while such an increase immediately followed by a unitary Qdot loss indicates full-collapse fusion[59]. In neurons on

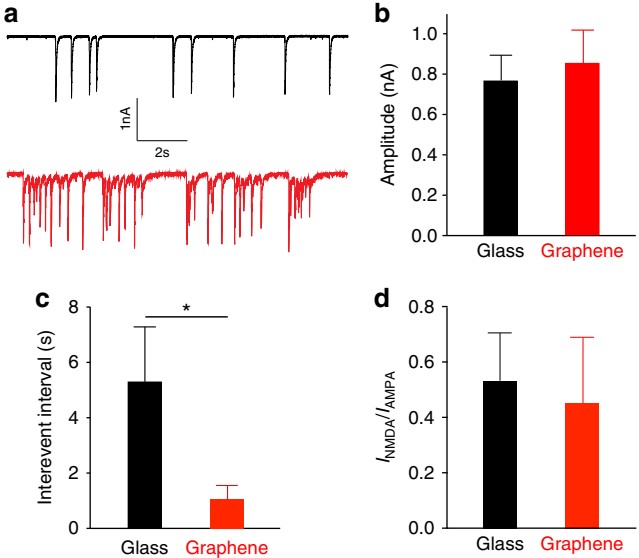

**Fig. 5** Graphene increases spontaneous firing frequency in neurons. **a** Sample traces from neurons on glass (black) or graphene (red). **b** Mean sEPSC amplitudes ($n_{glass}$ = 13 cells, $n_{graphene}$ = 15 cells, $N$ > 4 batches; $p$ > 0.1, two-tailed $t$-test, Cohen's $d$ = 0.10). **c** Mean sEPSC inter-event intervals ($n_{glass}$ = 13 cells, $n_{graphene}$ = 15 cells, $N$ > 4 batches, *$p$ < 0.05, two-tailed $t$-test, Cohen's $d$ = 1.30). **d** $I_{NMDAR}/I_{AMPAR}$ ($n_{glass}$ = 10 cells, $n_{graphene}$ = 9 cells, $N$ > 3 batches; $p$ > 0.1, two-tailed $t$-test, Cohen's $d$ = 0.27). Error bars are S.E.M.

graphene, individual synaptic vesicles conducted more FRF (Supplementary Figure 5d) and, correspondingly, a significantly larger fraction of FRF out of all vesicle release events (Fig. 7h), which would be expected for a cholesterol increase in the plasma and vesicular membranes of presynaptic terminals[32–35]. Thus we concluded that graphene promotes fast synaptic vesicle turnover in addition to increases in vesicle number and their release probability.

**Synaptic potentiation on graphene is cholesterol dependent.** To further investigate the extent to which membrane cholesterol levels mediate presynaptic potentiation on graphene, we sought to directly manipulate cell membrane cholesterol levels. We first increased membrane cholesterol levels in neurons grown on glass by the addition of TFC as an exogenous cholesterol supply[60]. Neurons were treated with 1 μM TFC for 1 h, which was sufficient to allow incorporation into the plasma membrane. We used two independent measurements to assess cholesterol levels: Filipin staining (again measured in neurites) (Fig. 8a, b) of fixed neurons and GP imaging (Fig. 8c) of live neurons. To avoid the cross-excitation of TFC when imaging C-laurdan, all GP imaging experiments involving the manipulation of membrane cholesterol levels were performed using a different optical configuration (see Methods and Supplementary Figure 7). Notably, although Filipin staining intensity was greater for the acute addition of TFC than for chronic growth on graphene, the similarity in GP distributions suggests that cholesterol levels within the plasma membrane were similar between graphene and TFC-treated glass samples. TFC treatment of neurons on glass increased the FM4-64 destaining rate, increased the FRF ratio, and also increased release probability (Fig. 8d, e and Supplementary Figure 5b&c), similar to the effects we observed on graphene. We next used methyl-β-cyclodextrin (MβCD), a cholesterol-binding

compound[61], to decrease cholesterol levels in neurons on graphene. To limit its effect to the cell membrane, we applied a low concentration (0.5 mM) of MβCD for a short time (10 min). For all four conditions, 10 μM D-AP5 and 5 μM NBQX were co-applied to prevent the activity-induced exposure of intracellular membrane cholesterol. Again, independent assessment of cholesterol levels by both Filipin staining in fixed neurites (Fig. 8a, b) and GP imaging of live neurons (Fig. 8c and Supplementary Figure 8) confirmed a reduction of cholesterol levels. We then imaged FM4-64 and Qdots unloading during high K+ stimulation. MβCD treatment decreased the rate of FM4-64 loss, the amount of fast vesicle fusion, and the vesicle release probability (Fig. 8d, e and Supplementary Figure 8b&c) of neurons on graphene, demonstrating that reversing the graphene-induced cholesterol increase via MβCD application also reverses graphene's effect on synaptic vesicles. Our data collectively demonstrate that cholesterol, most likely in the plasma membrane, is an important mediator of graphene's ability to potentiate neurotransmission.

**Cholesterol-dependent potentiation of P2YR signaling on graphene.** As membrane cholesterol plays an integral role in the binding and regulation of many transmembrane proteins[62], we asked if and how cholesterol enrichment on graphene substrates could affect transmembrane proteins and the signaling pathways they mediate. Using the same approach that was used for our neuronal culture configuration, mouse fibroblast cells (NIH 3T3) were plated directly on graphene or glass. Filipin staining demonstrated a ~49% increase of fluorescence intensity in 3T3 cells on graphene (Fig. 9a, b), a much greater increase than what was observed in neurons, possibly due to lower homeostatic concentrations of plasma membrane cholesterol in 3T3 cells[63] than in neurons[64]. GP imaging suggested a modest reduction of lipid membrane fluidity on graphene (Fig. 9c). Among many cholesterol-sensitive transmembrane proteins, we chose therapeutically valuable P2Y receptors (P2YRs)[65], a class of GPCR. GPCRs are one of the largest protein families in the human genome[66] and represent about half of all modern pharmaceutical targets[67]. Structural models have elucidated that membrane cholesterol allosterically promotes GPCR activity by binding to the transmembrane domain[25]. In 3T3 cells, P2YRs mediate a fast Ca$^{2+}$ response to extracellular ATP[67], which can be quantitatively measured at high spatiotemporal resolution via Ca$^{2+}$ imaging. We applied two ATP stimuli at a 1-min interval and observed that the second Ca$^{2+}$ response was significantly diminished (Fig. 9d). This is consistent with P2YR-mediated Ca$^{2+}$ release from internal stores, which require longer than 1 min to refill. We then pharmacologically isolated relevant components in the transmembrane signaling pathway. Application of PPADS (a selective P2YR antagonist) inhibited Ca$^{2+}$ responses, confirming P2YRs as the ATP receptor (Fig. 9d). Both Tharpsigargin (an agonist for Ca$^{2+}$ release from internal stores) and Ca$^{2+}$-free bath solution (preventing the refilling of internal Ca$^{2+}$ stores) reduced the second Ca$^{2+}$ response, collectively confirming internal stores as the Ca$^{2+}$ source (Fig. 9d).

We observed a significantly larger Ca$^{2+}$ response to the first ATP stimulus and, subsequently, a smaller response to the second stimulus in 3T3 cells on graphene relative to those on glass (Fig. 9e), demonstrating that P2YR-mediated Ca$^{2+}$ responses are enhanced. Since ATP stimulation was still required for Ca$^{2+}$ release, the facilitation we observed in cells on graphene was likely allosteric. In the same manner as the studies we performed using neurons, we next manipulated cholesterol to study its role in the enhanced Ca$^{2+}$ responses we observed on graphene. 3T3 cells growing on graphene or glass were pretreated with MβCD or TFC, respectively, using the same protocols we used for neurons.

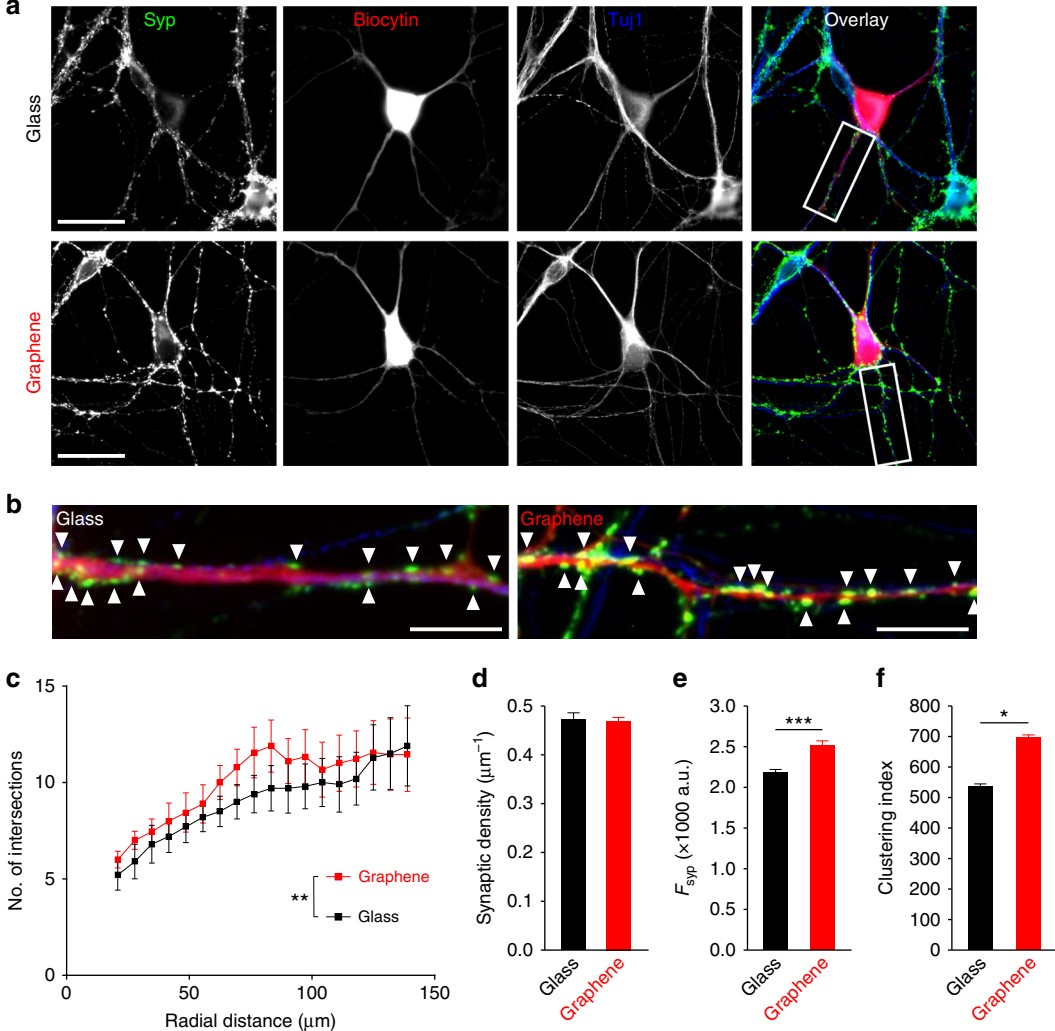

**Fig. 6** Morphological comparison of neurons on glass or graphene. **a** Sample images of immunofluorescence staining for the synaptic vesicle marker, Synaptophysin (Syp, green), and neuron-specific class III β-tubulin (TuJ1, blue) in recorded cells (biocytin filled, red). Scale bar, 30 μm. **b** Inset regions from the sample images are indicated by white boxes. Scale bar, 15 μm. Arrowheads indicate examples of Syp puncta. **c** Sholl analysis (**$p = 0.01$, $F(1,323) = 6.7$, two-way ANOVA with repeated measures followed by a Bonferroni multiple comparisons test, $\omega^2 = 0.017$, $n_{glass} = 12$ cells, $n_{graphene} = 11$ cells, $N > 4$ batches). **d** Lateral density of Syp puncta along the neurites of recorded neurons (Tuj1-positive and biocytin-positive) ($n_{glass} = 12$ cells, $n_{graphene} = 11$ cells, $N > 4$ batches; $p > 0.1$, two-tailed $t$-test). **e** Average Syp immunostaining ($n > 10,000$ synapses analyzed, $N > 3$ batches, ***$p < 0.001$, two-tailed $t$-test, Cohen's $d = 0.500$). **f** Average Syp clustering within individual synaptic boutons ($n \geq 3$ FOVs, $N = 3$ batches; *$p < 0.05$, two-tailed $t$-test, Cohen's $d = 0.65$). Error bars are S.E.M.

TFC pretreatment increased Filipin staining intensity in 3T3 cells on glass (~15%, Fig. 9a, b), although not to the levels observed on graphene, and membrane fluidity was moderately reduced (Fig. 9c). Conversely, MβCD treatment of 3T3 cells on graphene reduced Filipin staining intensity (Fig. 9a, b) and increased membrane fluidity (Fig. 9c). Although we did observe differences in Filipin intensity when comparing non-treatment vs. treatment conditions, our GP data demonstrate that our treatments were consistent in altering plasma membrane cholesterol in that MβCD application resulted in GP distributions similar to glass (Fig. 9c) and TFC application resulted in GP distributions similar to those on graphene. Our data are consistent with the idea that membrane cholesterol levels are increased within a certain physiological range, but over the length of our 3T3 cell culture, additional cholesterol may be trafficked to and distributed homogeneously in intracellular membrane areas. Again, our bidirectional manipulations of cholesterol resulted in functional outcomes similar to what we observed for neurons on glass or

graphene (Fig. 9e). Enhanced $Ca^{2+}$ responses on graphene were significantly diminished by MβCD treatment, and $Ca^{2+}$ responses on glass were significantly potentiated by TFC application (Fig. 9e). Together, these results suggest that a graphene-induced cholesterol increase is capable of potentiating cell signaling pathways via transmembrane proteins whose activities are allosterically regulated by cholesterol.

## Discussion

Here we show that pristine monolayer graphene, when in chronic contact with the cell membrane, can modify cellular processes via increased cholesterol. In cell-free systems, we observed that graphene extracts cholesterol from cell culture media and quenches a fluorescent cholesterol analog, consistent with prior predictions of a graphene–cholesterol interaction. Cell-based measurements cooperatively revealed that graphene increased plasma membrane cholesterol. In neurons, this results in a presynaptic potentiation

of neurotransmission, realized by increases in synaptic vesicle number, release probability, and turnover rate. Notably, all of these are regulated by membrane cholesterol[32–35,56]. Manipulation of membrane cholesterol levels validated the correlation between graphene-induced membrane cholesterol increase and changes in synaptic vesicle number and behavior. We extended our findings by studying graphene's impact on integral membrane receptors that are known to be affected by cholesterol

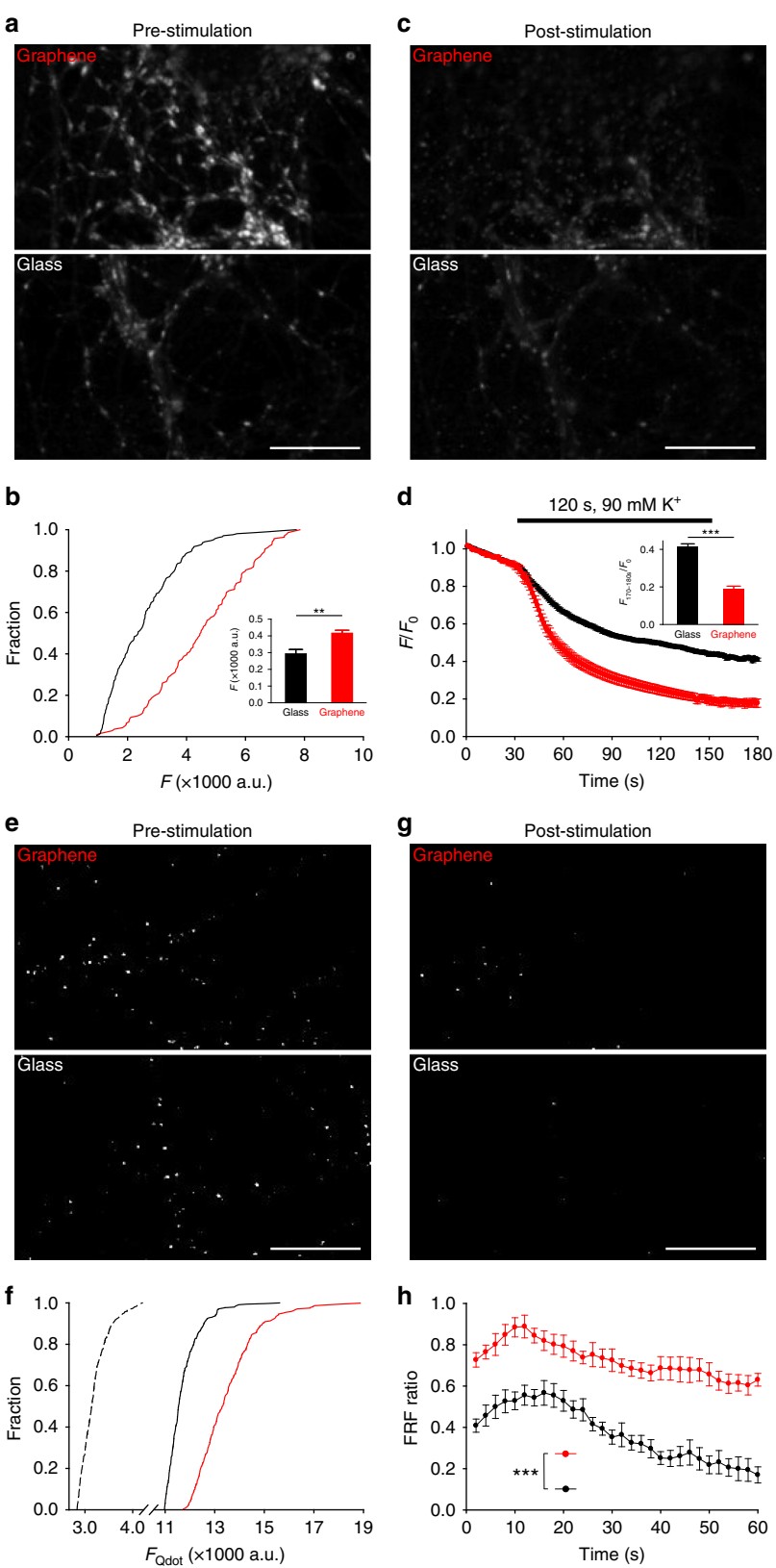

concentration and observe enhanced P2YR-mediated $Ca^{2+}$ responses in fibroblast cells on graphene. Congruent with our findings in neurons, we demonstrate that this potentiation is facilitated by cholesterol.

Both LPE and CVD produce single- and few-layer graphene with similar surface characteristics[2,36,42,43]. Thus the interaction demonstrated using LPE GFs should remain valid for the CVD graphene films used in our cell-based studies. The nature of this interaction likely involves both a hydrophobic interaction and stacking between cholesterol's planar tetracyclic ring group and graphene's hexagonal lattice[27–31]. Our spectral data (Fig. 1b and Supplementary Figure 1) suggest that the interaction may be more complex than a simple hydrophobic interaction. Further work, including both empirical studies and computational modeling, will aid in elucidating the mechanisms of this interaction within complex membranes that contain many types of molecules. This will clarify the relative selectivity of cholesterol in comparison to other types of biomolecules, including phospholipids, peptides, and carbohydrates, which are common constituents of culture media and also located on the cell surface.

Given the importance of minimizing the distance between graphene and cells, we chose an approach similar to previous studies[17,19]—cells were plated directly on bare CVD graphene films without a (e.g.) Matrigel coating. Because it gels upon incubation at physiological temperatures, Matrigel acts as an effective interlayer between the substrate and the cell membrane in traditional culture configurations. We reasoned that removal of this layer would better resemble scenarios where graphene is used as a biosensor. However, this approach does not prevent biomolecule deposition from culture media onto the graphene surface. A protein corona is likely formed over the length of our culture period. Hu et al.[68] recently demonstrated that within minutes of exposure to serum-containing media a stable protein corona will form on graphene oxide, which has a more favorable surface for protein adsorption than graphene. We have found a similar time course for dissociated neurons to attach to bare glass surfaces (Supplementary Figure 6). This suggests that as cells are adhering to graphene there may still be areas where protein adsorption does not completely cover the surface, which would allow for contact between graphene and the cell membrane. Intriguingly, when neurons were cultured in conditioned media, previous reports have demonstrated that the crystallinity of the graphene surface is a determinant of axon outgrowth, suggesting that even with biomolecule deposition, graphene still acts to alter cell function[17]. This is in line with the findings of our cholesterol assay (Fig. 1a), where chronic exposure resulted in cholesterol enrichment on the graphene surface. The development of biocompatible surface modification strategies to minimize fouling on carbon nanomaterials remains an important issue for improving biosensor lifetime, but our results demonstrate that, even in the presence of a biomolecular corona, cholesterol is a mediator of graphene's functional effects.

We employed two independent approaches to evaluate changes in cellular cholesterol levels on graphene substrates: Filipin staining and GP imaging. Filipin fluoresces upon cholesterol binding and is a well-accepted qualitative reporter for cellular cholesterol in fixed cells. It permeabilizes the cell membrane and binds intracellular cholesterol as well as other lipids[45]. This may explain the inconsistency in the absolute difference between graphene and glass (Fig. 3b vs. Fig. 8b); notably the overall increase on graphene was substantial across experiments. To obtain an additional measure of cholesterol that would not disturb the membrane, we employed GP imaging in live cells. The magnitudes of GP distribution shifts across different experiment sets were consistent (~0.15). Although much smaller than the relative changes we observed for Filipin staining, such changes are in line with what has been observed previously for GP value at different cholesterol amounts[69]: modest absolute changes correlate to much larger differences in overall cholesterol content. To empirically demonstrate how GP shifts represent differences in neuronal membrane cholesterol, we performed GP imaging on neurons after weak or strong cholesterol depletion by MβCD (Supplementary Figure 8)[61]. Because the GP changes we observed on graphene or after bidirectional manipulations (~ 0.15) lie between the shifts we observed for weak (~0.089) and strong (~0.191) MβCD treatment, we conclude that the membrane cholesterol changes were moderate. Broadly, both GP and Filipin were qualitatively consistent with the observation that cells on graphene had increased TFC labeling (Fig. 4), all of which support our conclusion that graphene increases cholesterol. However, because TFC acts as an exogenous source of cholesterol and was applied acutely, quantitative comparison of TFC labeling to Filipin staining or GP shift is challenging.

After the addition or removal of cholesterol from neurons on glass or graphene respectively, Filipin staining seemingly reported large changes in cellular cholesterol (Fig. 8b), whereas neither neuronal membrane rigidity (Fig. 8c) nor synaptic vesicle release (Fig. 8d, e) reported changes exceeding those on glass or graphene. We speculate that, although cholesterol levels can be increased, there are additional factors that place some limit on cholesterol's ability to modulate cellular function. For example, membrane rigidity also requires the involvement of saturated or unsaturated fatty acids to maintain or modify lipid phase order. And for synaptic vesicles, changes in number were likely confined by the sizes of the different vesicle pools. As only a subpopulation of synaptic vesicles is releasable, this may explain why Syp staining, which labels all vesicles, exhibited a smaller increase (~21%) in the graphene group than FM or Qdot loading, which only label releasable vesicles (~26–30%). Furthermore, the ability of excess cholesterol to be incorporated into fusion machinery as well as the limited number of release sites may help to set bounds beyond which overall homeostasis would be irreparably disrupted. We also cannot exclude the possibility that Filipin labels

**Fig. 7** Graphene induces presynaptic potentiation. **a** Sample images of FM1-43 labeling. Scale bar, 30 μm. **b** Cumulative distributions of FM1-43 intensities at synaptic boutons (black, glass; red, graphene, same color coding hereafter). $n_{glass} = 207$ ROIs, $n_{graphene} = 139$ ROIs, $N = 3$; $p < 0.05$, Kolmogorov–Smirnov test). Inset. Average FM1-43 fluorescence. ** $p < 0.01$, two-tailed $t$-test. **c** Sample images of FM1-43 labeling after destaining. Scale bar, 30 μm. **d** FM1-43 fluorescence during destaining. Inset is average fluorescence from 170 to 180 s ($n_{glass} = 207$ ROIs, $n_{graphene} = 139$ ROIs, $N = 3$; *** $p < 0.001$, two-tailed $t$-test). **e** Sample images of single Qdot loading. Scale bar, 30 μm. **f** Cumulative distributions of Qdot intensity after background subtraction in ROIs defined by retrospective FM4-64 labeling (single Qdot loading, dotted line; total recycling pool loading, solid lines). The average single Qdot intensity after background subtraction is 378 ± 41 a.u. The average total Qdot intensities after background subtraction are 8787 ± 156 a.u. for glass and 11,050 ± 224 a.u. for graphene ($n_{glass} = 187$ ROIs, $n_{graphene} = 211$ ROIs, $N = 4$; $p < 0.001$, Kolmogorov–Smirnov test). The estimated average numbers of total recycling vesicles are 23.2 for glass and 29.2 for graphene. **g** Sample images of single Qdot labeling after stimulation. Scale bar, 30 μm. **h** Fast-and-reversible fusion (FRF) ratio (out of all fusion events) during 1-min 10-Hz field stimulation ($n_{glass} = 174$ ROIs, $n_{graphene} = 181$ ROIs, $N = 3$; *** $p < 0.001$, two-tailed $t$-test on the average FRF values from a five-frame window at the end of each time course). Error bars are S.E.M.

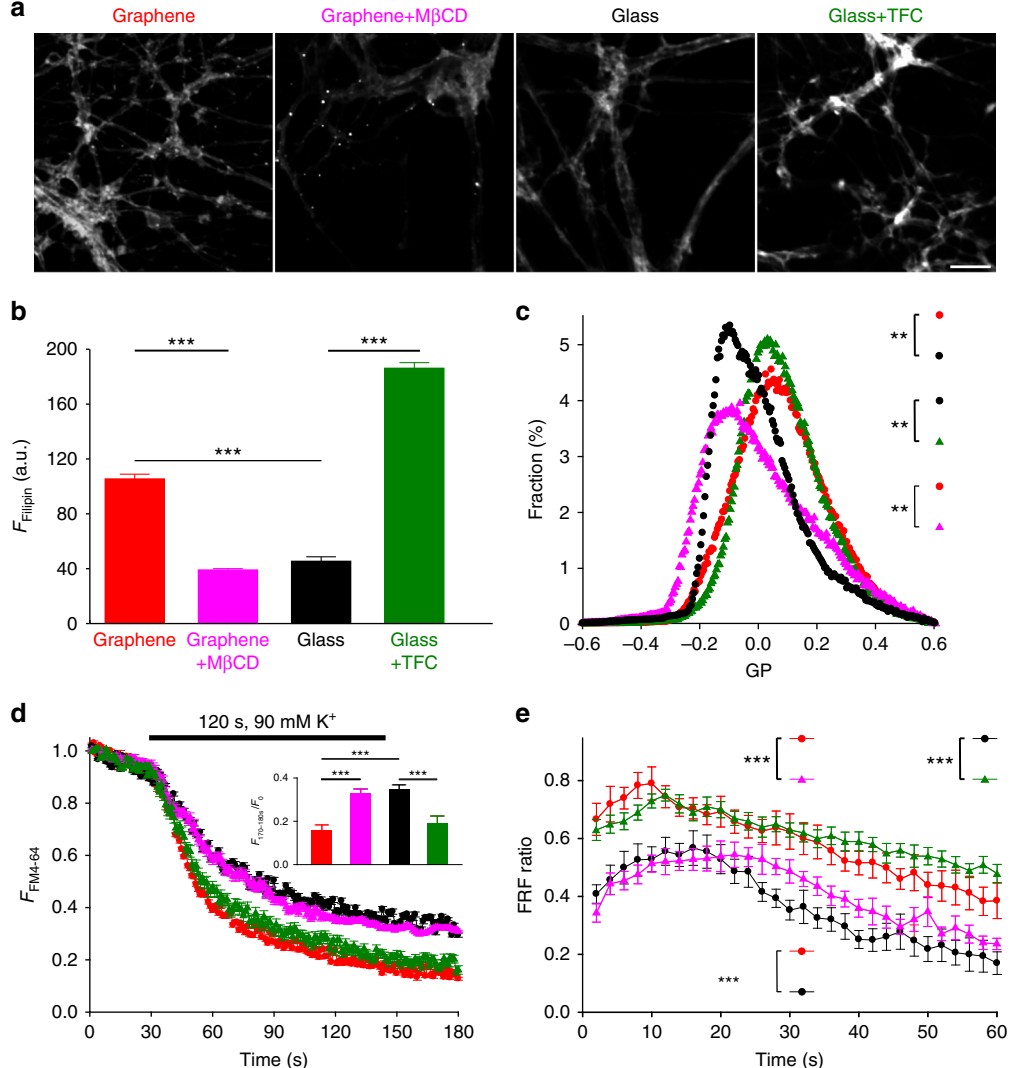

**Fig. 8** Cholesterol mediates graphene-induced presynaptic changes. **a** Sample Filipin staining images of neurites on graphene with (purple) or without (red) MβCD treatment and on glass with (green) or without (black) TFC loading. Scale bar, 50 μm. Same color coding hereafter. **b** Average Filipin staining intensities in neurites ($n_{graphene} = 107$ neurites, $n_{graphene+MβCD} = 151$ neurites, $n_{glass} = 188$ neurites, $n_{glass+TFC} = 152$ neurites, $N = 3$ batches for every group; for graphene vs. graphene+MβCD, graphene vs. glass, and glass vs. glass+TFC, ***$p < 0.001$, all Wilcoxin rank-sum tests). **c** Distributions of GP values over individual image pixels ($n = 6$ FOVs, $N = 3$ batches for every group; for graphene vs. graphene+MβCD and glass vs. glass+TFC, both **$p < 0.01$, for graphene vs. glass+TFC, $p = 0.073$, Kolmogorov–Smirnov test of the distributions of GP values, see 'Statistical analysis' section in Methods). **d** FM 4-64 fluorescence changes before and during 2-min 90-mM K$^+$ and (inset) average fluorescence decrease using a five-frame window at the end of the stimulation period ($n = 6$ FOVs, $N = 3$ batches per group; for graphene vs. graphene+MβCD, graphene vs. glass, and glass vs. glass+TFC, ***$p < 0.001$, for graphene vs. glass+TFC and graphene+MβCD vs. glass, N.S. $p > 0.05$, all two-tailed $t$-tests). **e** FRF ratios during 1-min 30-Hz electrical stimulation ($n = 3$ FOVs, $N = 3$; for graphene vs. graphene+MβCD and glass vs. glass+TFC, both ***$p < 0.05$, two-tailed $t$-tests on the average of a five-frame window at the end of the stimulation period). Error bars are S.E.M.

intracellular cholesterol, which may be affected by our manipulations but is not accounted for in GP or FM/Qdot imaging.

After bidirectional manipulation of cholesterol levels, we observed that the resulting changes in cellular cholesterol and membrane rigidity were larger in neurons than in 3T3 cells (Fig. 8b vs. Fig. 9b and Fig. 8c vs. Fig. 9c). This discrepancy may be due to metabolic and homeostatic differences in membrane cholesterol between neurons and 3T3 cells[63,64,70]. Moreover, neurons have higher levels of plasma membrane cholesterol[64] than 3T3 cells[63] and cells may respond differently to MβCD or TFC depending upon both concentration and treatment time.

We have demonstrated that neurons grow and form functional synapses on both glass and monolayer graphene substrates without significant developmental or gross morphological defects. This is seemingly different from a recent report[17], which focused on neuronal development and noted that neurons were unable to grow normally on bare glass. There are several technical differences that may explain this discrepancy, including species and age of cells at plating. The use of serum in our culture media serves as a rich source of cholesterol in comparison to the reported study[17] where serum-free conditioned media was used. Our culture media contains approximately 12–14 μM cholesterol (Fig. 1a), whereas astrocyte conditioning only further increases this concentration

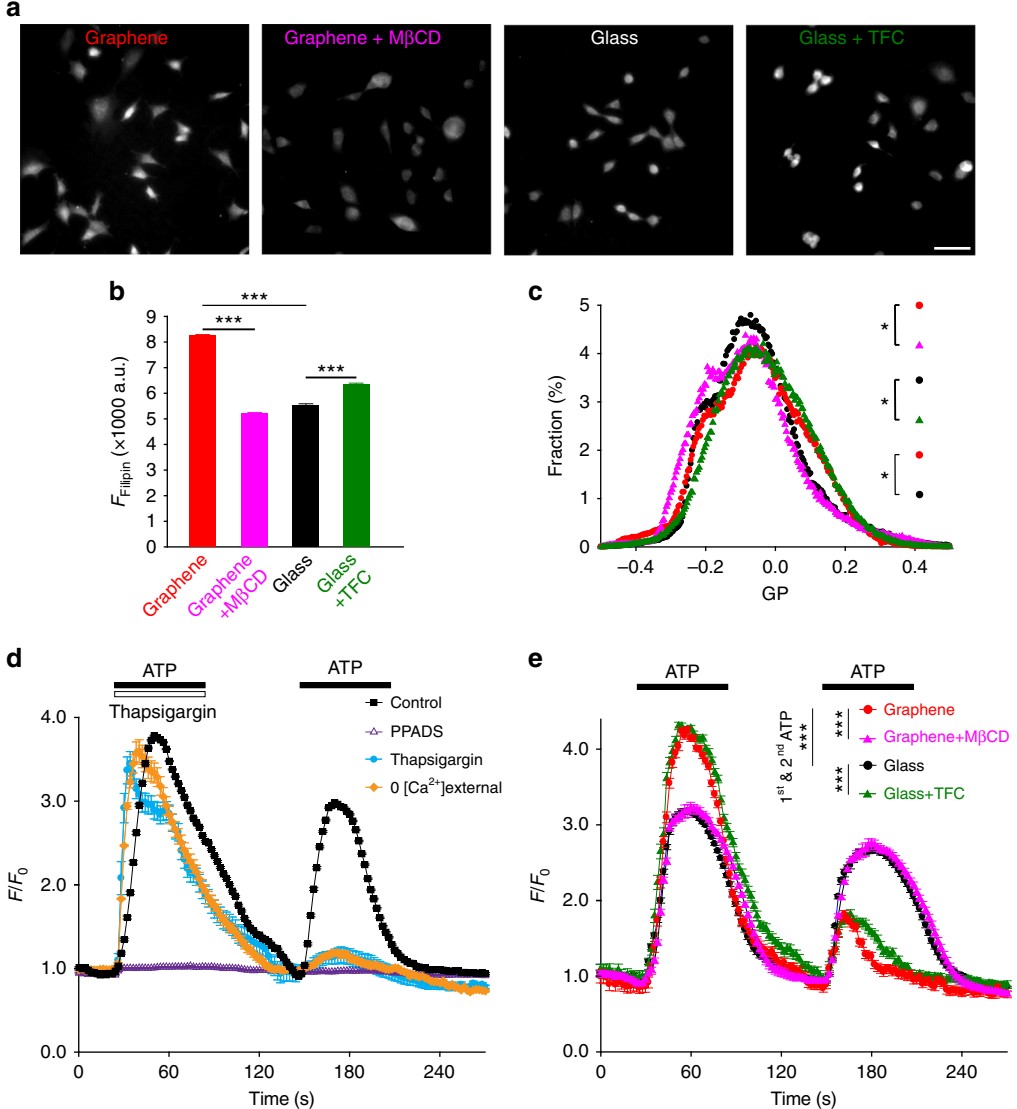

**Fig. 9** Graphene enhances P2Y receptor-mediated $Ca^{2+}$ responses. **a** Sample images of Filipin staining of 3T3 cells on graphene with (purple) or without (red) MβCD treatment and on glass with (green) or without (black) TFC loading. Same color coding hereafter. Scale bar, 50 μm. **b** Average intensities of Filipin staining ($n_{graphene} = 1536$ cells, $n_{graphene+MBCD} = 2286$ cells, $n_{glass} = 1317$ cells, $n_{glass+TFC} = 1487$ cells, $N = 3$ batches for every group; for graphene vs. graphene+MβCD, graphene vs. glass, and glass vs. glass+TFC, ***$p < 0.001$, all Wilcoxin rank-sum tests). **c** Distributions of GP values over individual image pixels ($n = 6$ FOVs, $N = 3$ batches; for graphene vs. graphene+MβCD graphene vs. glass, glass vs. glass+TFC, all *$p < 0.05$, for graphene vs. glass +TFC, $p > 0.05$, Kolmogorov–Smirnov test of the distributions of GP values, see 'Statistical analysis' in the Methods section). **d** Two consecutive 100-μM ATP applications elicited the release of $Ca^{2+}$ from internal $Ca^{2+}$ stores ($n = 6$ FOVs, $N = 3$ batches for every condition). The second $Ca^{2+}$ response was smaller than the first with a 1 min interval between. Both $Ca^{2+}$ responses were blocked by 50 μM PPADS (pyridoxalphosphate-6-azophenyl-2′,4′-disulphonic acid), a P2YR inhibitor (white triangles). Thapsigargin 1 μM (blue dots) elicited a similar $Ca^{2+}$ response as ATP but significantly reduced the second response by exhausting internal $Ca^{2+}$ stores. In the absence of extracellular $Ca^{2+}$ (the source for refilling internal $Ca^{2+}$ stores) (orange diamonds), the second response was also significantly reduced ($n = 6$ FOVs, $N = 3$ batches for every condition). **e** ATP-elicited $Ca^{2+}$ release from internal stores was facilitated by graphene or TFC pretreatment and reduced by MβCD ($n = 6$ FOVs, $N = 3$ batches for every group; for graphene vs. graphene+MβCD, graphene vs. glass, and glass vs. glass+TFC, ***$p < 0.001$, two-tailed $t$-tests). Error bars are S.E.M.

by 2 μM (Fig. 1a). The additional growth and attachment factors present in serum likely help to mitigate any deficits in neuronal adhesion and development. Functionally, we observe no difference in neuronal membrane conductance (similar sEPSC amplitudes, Fig. 5) and no $Ca^{2+}$ leakage in 3T3 cells (stable basal cytosolic $Ca^{2+}$ concentration without stimulation or after P2YR inhibition, Fig. 9d) on graphene, extending the findings of previous studies which have demonstrated that graphene substrates do not damage the eukaryotic membrane[9–11]. TFC and FM dye

labeling further indicate that membrane integrity and trafficking are uncompromised, as labeling would have appeared atypical with significant membrane degradation.

There are two scenarios that explain the enrichment of cholesterol we observe. Chronic contact between graphene and cells may attract cholesterol to areas nearest to the graphene surface, reducing cholesterol in other membrane compartments such as the endoplasmic reticulum (where cholesterol sensors reside[71]). Consequently, cholesterol synthesis and/or uptake is upregulated,

resulting in a total cholesterol increase over time (Figs. 3, 8a and 9a). An alternative scenario seems equally possible: that graphene adsorbs cholesterol from the culture media onto its surface (Fig. 1a), providing an enriched local cholesterol supply to the neighboring plasma membrane. Membrane cholesterol levels are elevated and over time excess cholesterol is trafficked to internal membranes (Fig. 9a–c). This is in line with evidence demonstrating that the uptake of exogenous cholesterol is critical for mature neuronal function[56]. Differentiating between these two scenarios will require the acute application of graphene and the ability to monitor cholesterol trafficking in live cells at adequate spatiotemporal resolution.

Although graphene has been documented to damage bacterial cell membranes[13], it has seemingly few adverse effects on eukaryotic cells based on our own and others' previous observations[17–19]. This difference between cell types can be explained in part by our cell culture configuration and in part by the presence or absence of membrane cholesterol. Prokaryotic membrane destruction was caused by nanometer-sized GFs that pierced through the bacterial membrane causing lipids to disperse[13]. In our and an additional study[17], large CVD graphene films are essentially fixed on a surface (Fig. 2). The extremely high in-plane strength of graphene effectively prohibits small flakes from breaking off of the glass surface and inserting into the cell membrane. Therefore, graphene remained parallel to the cell membrane during culture and was unlikely to cause membrane disruption in the same manner as previously described. Fully distinguishing the role cholesterol plays in the cellular responses to different types of graphene will require further study using both GFs applied to eukaryotic cells and chronic growth on graphene of prokaryotic cells.

Cholesterol modulates many membrane-associated proteins and other cellular functions in addition to its role in the structural integrity of the eukaryotic plasma membrane. Thus it would be interesting to further study graphene's impact on other signaling pathways regulated by cholesterol. For example, it helps to define the fluidity of the lipid bilayer, which in turn regulates the dynamics and subcellular distribution of many transmembrane proteins[24]. Cholesterol is also essential for the activity of various disease-associated membrane proteins like γ-secretase, an amyloidogenic enzyme thought to be an important therapeutic target in Alzheimer's disease. In addition, cholesterol is a key constituent of membrane nanodomains (a.k.a. lipid rafts), which are believed to act as the nexus for transmembrane protein complexes, mediating signal transduction across the plasma membrane, carrying out receptor-mediated endocytosis and more[72]. Therefore, further study is warranted for a comprehensive view of graphene's effect on the organization and trafficking of eukaryotic cell membranes as well as membrane protein distribution and mobility.

For the majority of graphene-based bioapplications, especially those seeking to harness its unique electrical properties, cells or tissue need to be directly interfaced with the surface to maximize detection efficiency. Given our findings, these applications should further consider the involvement of the cell membrane when evaluating the effect of the material on any biological system. The effects we observe may also occur in other carbon allotropes, for example, carbon nanotubes. It is possible that the variation of atomic structure among different allotropes may uniquely influence their cholesterol affinity; thus investigating the shared or unique effects of other carbon allotropes on the plasma membrane and whether cholesterol is involved in these effects will be highly informative. New advances in the ability to modify carbon nanomaterials without compromising their electrical properties may help to tailor interactions with cholesterol or other biomolecules. Since cholesterol is a precursor for many steroids, which also contain the same tetracyclic ring, graphene could potentially be utilized to detect, deliver, or manipulate steroids in vitro and in vivo. This opens future directions for graphene in biomedicine but also demands further structural and mechanistic investigation of the membrane interaction between graphene and diverse biomolecules.

## Methods

**Graphene production.** To produce GFs, we used liquid exfoliation of graphite powder (ASBURY CARBONS, Grade: 2299) in the presence of 2 wt% PVP (molecular weight: 1,300,000 g mol$^{-1}$, from Sigma) water solution and sonicated for 9 h in a bath sonicator. The uniform GF suspension was then centrifuged with a Thermo Scientific Fiberlite F15-6 X 100y rotor at 4000 r.p.m. and at room temperature for 1 h to sediment large graphite aggregates. The upper 50% of supernatant was carefully decanted, resulting in PVP-functionalized GF suspension[36]. The transmission characterization of GF suspension was carried out on a Varian Cary 5000 UV-VIS-NIR spectrophotometer. The concentration of GF was estimated with an absorption coefficient of 2460 L/g/m at 660 nm[36], which is typically 26 mg L$^{-1}$ for freshly made GF suspension. The suspension is stored at 4 °C and remains stable for >3 years. Single-layer graphene sheets were synthesized by CVD on a copper foil with 100 sccm (standard cubic centimeter per minute) hydrogen and 10 sccm methane as the feed gases[42]. A layer of poly(methyl methacrylate) (PMMA) was then spin-coated onto the graphene film that was grown on the copper foil, which was later removed through wet etching in ferric chloride solution. Subsequently, the graphene film was transferred onto glass coverslips, and the PMMA support was dissolved in acetone.

**Raman spectroscopic characterization of graphene.** The quality of graphene sheets was examined by a DXR Raman Microscope (Thermo Scientific). A 532-nm laser (~5 mW power) was expanded and focused to a diffraction-limited laser spot (<1 μm) through a 50× Olympus objective. The intensity features of graphene Raman spectra could be found at ~1590 and ~2680 cm$^{-1}$, corresponding to the G and 2D modes, respectively. The high 2D-to-G intensity ratio (>1) and symmetric shape of the 2D peak indicate that the graphene is monolayer. For Raman mapping, the samples were moved by a motorized microscope stage at a step of 1 μm for both $x$ and $y$ axis. A Raman spectrum was recorded at each position, and the intensities of the G and 2D modes were plotted to form spatially resolved images.

**Cholesterol assay.** An enzymatic assay (Amplex Red Cholesterol Assay Kit, Life Technologies) was used to quantify free cholesterol concentrations according to the manufacturer's instructions[37]. Briefly, a serial dilution of cholesterol standard (0, 2, 5, 10, 15 and 20 μM) was used to generate a calibration curve. One-milliliter aliquots of media were incubated as prepared or with the addition of 0.002 wt% PVP in phosphate-buffered saline (PBS), or 260 ng mL$^{-1}$ graphene nanoflakes (GFs) with 0.002 wt% PVP in PBS at 37 °C and 5% CO$_2$ for 24 h. Graphene fractions were separated from media using Amicon centrifugal filters (100 kDa, Millipore) and resuspended in 1 mL of fresh media. The flow-through media was also collected for the graphene fraction. Media alone, media incubated with PVP, GNFs separated from media, and flow-through media were assayed. The fluorescence intensities of all enzymatic reaction products were measured using a spectrofluorometer (FluoroMax-4, Horiba). Excitation and emission slit widths were set at 5 nm for all measurements. Samples were excited at 570 nm, and emission was measured at 590 nm. For each group, four different batches of media were used, and four independent measurements were performed for each media sample. Cholesterol concentrations were then calculated using the standard curve.

**Spectrofluorometer measurements.** TFC (Avanti) or BODIPY (Thermo Fisher) were diluted in water at a final concentration of 1.3 μM and incubated at room temperature for all time points. All fluorescence emission measurements were performed using a FluoroMax-4 spectrofluorometer (Horiba). Excitation and emission slit widths were set at 5 nm for all measurements. Samples were excited at 400 nm to avoid spectral bleed-through and non-specific excitation in mixed solutions. Emission spectra were collected from 500 to 650 nm. Three replicates were scanned three times each for every sample and the result averaged. Subsequently, averaged intensities were corrected for the broadband absorbance of either graphene or PVP across the emission spectra, as defined by:

$$F_{TFC,\lambda} + (1 - T)F_{TFC,\lambda},$$ where $F$ is the fluorescence emission value at a wavelength for TFC and $T$ is the transmittance value at that same wavelength for graphene or PVP calculated from the Beer–Lambert relationship. Data smoothing was performed using a nine-point Savitzky–Golay filter in Matlab.

**Cell culture.** All animal procedures and all experimental procedures were approved by the Vanderbilt University Animal Care and Use Committee (VUACUC, #M1500052) and were performed in accordance with the VUACUC

approved guidelines and regulations. Rat hippocampal cultures were prepared as previously described[44]. Hippocampal neurons (CA1–CA3) derived from postnatal (P0/P1) Sprague–Dawley rats of both sexes were used. Neurons were dissociated to a single-cell suspension, recovered by centrifugation, and resuspended in plating media composed of Minimal Essential Medium (MEM, Life Technologies) containing (in mM) 27 glucose, 2.4 NaHCO$_3$, 0.00125 transferrin, 2 L-glutamine, 0.0043 insulin, and 10%/vol fetal bovine serum (FBS, Omega). Because of the addition of FBS, all neuronal media contained cholesterol. Cell resuspension was deposited on round 12 mm coverslips at a density of ~200,000 cells/mL for two different surface conditions: bare glass or graphene-coated glass. After 2 h, 1 mL of culture media was added, a 1:1 mixture of plating and 4-Arac-containing media: MEM containing (in mM) 27 glucose, 2.4 NaHCO$_3$, 0.00125 transferrin, 1.25 L-glutamine, 0.0022 insulin, 2 Ara-C, 1 %/vol B27 supplement (Life Technologies), and 7.5 %/vol FBS. Ara-C mimimized astroglia proliferation. For cell attachment studies, coverslips were washed three times with Hank's solution and cells remaining in randomly selected fields of view were counted. Sister cultures supplying conditioned media were prepared in the same manner, except that 1 mL of plating media was added 4 h after plating to allow glial cell proliferation. After the confluence of glial cells (~2 DIV), 1 mL of 4-Arac media was added. For cells growing on graphene and glass, after 2 DIV, 1 mL of culture media was replaced with an equal volume of conditioned media from sister cultures. Experiments were performed on cultures at 3 and 7 DIV for developmental studies and between 13 and 17 DIV for all other studies using neurons.

NIH-3T3 cells were grown at 37 °C with 5% CO$_2$ in Dulbecco's modified Eagle's medium containing 4.5 g L$^{-1}$ glucose and l-glutamine supplemented with 10% FBS, 100 units/mL penicillin, and 100 µg mL$^{-1}$ streptomycin. Because of the addition of FBS, this media also contains cholesterol. Cells were regularly passaged to maintain adequate growth and were passaged at least five times before trypsinization and plating on either graphene-coated or bare glass coverslips (25 µL or ~2 × 10$^6$ cells per coverslip). Cells were grown to 50–80% confluence for 24 h on coverslips prior to imaging.

**Filipin staining and image analysis**. Cells were fixed in PBS containing 4% paraformaldehyde for 30 min, washed, and incubated with filipin (1:500 in PBS, Sigma-Aldrich) for 2 h at room temperature. Fluorescence imaging was performed on an Olympus IX-81 inverted microscope using a Nikon Intensilight illuminator, a Nikon Plan Apo VC 20× objective (N.A. 0.75) and a fluorescence filter set (Ex 390/40, DiC T425LPXR, Em 460/50, Semrock). Images were acquired with a CoolSnap K4 CCD camera (Photometrics) via Micro-manager with the same acquisition settings (exposure time = 300 ms and gain = 1) across each experimental group. For analysis, three independent batches of cultures were analyzed (n > 9 different coverslips). The total number of neurites or cells analyzed are reported (see figure legends). For the analysis of neurons, we manually selected ROIs covering neurites (Figs. 3 and 8) of morphologically identified neurons. Neurite selections were drawn to be approximately the same length for each ROI. For analysis of 3T3 cells, we used a threshold-based approach in ImageJ with a common threshold setting for all images in all experimental groups to select ROIs corresponding to cells. Average ROI intensity was measured in ImageJ. For every field of view (FOV), at least three ROIs from cell-free regions were manually selected, and their mean fluorescence intensities were calculated in the same manner. For background subtraction, the mean intensity value of every cell-containing ROI was subtracted by the average intensity of the three background ROIs in the same image.

**Immunocytochemistry and image analysis**. For immunostaining after electrophysiological recordings, coverslips containing recorded cells were fixed in PBS containing 4% paraformaldehyde, washed, blocked for 1 h with PBS containing 1% BSA, and incubated overnight at 4 °C with diluted primary antibodies (TuJ1—1:500, Synaptophysin—1:2000, and Streptavidin—1:500, all from Synaptic Systems). Secondary antibodies with distinct fluorophores (Alexa 488, 568 and 647, 1: 500 dilution for all, Biotium) were then incubated at room temperature for 2 h. Fluorescence imaging was performed on an Olympus IX-51 inverted microscope with a 60× UPlanFL (N.A. = 1.25) objective and a Flash 4.0 sCMOS camera (Hamamatsu). The optical filter sets (Chroma and Semrock) for Alexa 488, 568, and 647 fluorescence were, respectively: Ex 470/20 DiC 510LP 535/25, Ex 565/25 DiC 585LP Em 630/90, and Ex 630/60 DiC 660LP Em 695/100. For each fluorescence channel, images were taken with the same acquisition settings (excitation light intensity and exposure time: 500 ms for Alexa 488 anti-mouse, 500 ms for CF568 Streptavidin, 50 ms for Alexa 647 anti-guinea pig). Biocytin-positive neurons were positioned approximately in the center of the fields of view for imaging. Not all electrophysiologically recorded neurons were identified and imaged, as some were damaged during Biocytin infusion.

For Syp intensity analysis, a set of overlapping masks were used to restrict analysis to synapses on the processes of cells that were recorded (biocytin+). Binary masks of biocytin+ cells were generated by intensity-based global thresholding in ImageJ using a common threshold setting for the minimum value. Syp images corresponding to the same FOV were also thresholded to generate a second binary mask. Using the Boolean logic function in ImageJ, a new mask was generated from the intersection of the Syn+ and biocytin+ masks (AND function).

This mask was then subjected to minimum particle size restrictions and ROI sets were generated for each FOV. These ROI sets were applied to the original Syp image and mean intensities were measured. For cluster analysis, the standard deviation of the values obtained from the Syp intensity measurement were used. Background subtraction was performed on intensity values measured from each FOV by subtraction of the average of at least three different manually selected ROIs from cell-free areas of the Syp channel. Data were pooled for analysis, and the total number of synapses analyzed for each condition are reported in the figure legends.

**Sholl analysis**. Using the immunolabeled streptavidin-filled neurons from above, Sholl analysis was performed on images taken on an Olympus IX-51 inverted microscope with a 60× UPlanFL (N.A. = 1.25) objective and a Flash 4.0 sCMOS camera (Hamamatsu). The center of the filled soma was marked as the starting point and concentric circles spaced equidistantly were drawn. At each distance, the number of intersections was manually counted. Manual counting was taken as the best ground truth[73] to exclude possible artifacts from image thresholding and further to ensure that all counted projections originated from the cell of interest. Error bars represent the S.E.M. of the pooled intersections from all cells in a treatment group at each distance.

**Electrophysiology**. At least three coverslips from five batches of neuronal culture were selected for recording, using at least one neuron per coverslip. Whole-cell voltage clamp recordings were performed on 13–17-DIV neurons using a Multi-Clamp 700B amplifier, digitized through a Digidata 1440 A, and interfaced via pCLAMP 10 (all from Molecular Devices). All recordings were performed at room temperature. Cells were voltage clamped at −70 mV for all experiments. Patch pipettes were pulled from borosilicate glass capillaries with resistances ranging from 3 to 6 MΩ when filled with pipette solution. The bath solution (Tyrode's saline) contained (in mM): 150 NaCl, 4 KCl, 2 MgCl$_2$, 2 CaCl$_2$, 10 N-2 hydroxyethyl piperazine-n-2 ethanesulphonic acid (HEPES), 10 glucose, pH 7.35. The pipette solution contained (in mM): 120 Cesium Methanesulfonate, 8 CsCl, 1 MgCl$_2$, 10 HEPES, 0.4 ethylene glycol-bis-(aminoethyl ethane)-N,N,N',N'-tetraacetic acid (EGTA), 2 MgATP, 0.3 GTP-Tris, 10 phosphocreatine, QX-314 (50 µM), 5 biocytin (Tocris), pH 7.2. For the recordings of mEPSCs, bath solution was supplied with 1 µM tetrodotoxin (TTX, Abcam). The last 50 mEPSCs at the end of 5 min recordings in the presence of TTX were collected and analyzed using template-based event detection. The template was generated from our own representative data. To measure AMPA receptor currents, 20 µM D-(-)-2-Amino-5-phosphonopentanoic acid (D-AP5, Abcam), an NMDA receptor antagonist, was added to the bath solution. NMDA receptor currents were recorded in the presence of 10 µM 2,3-dihydroxy-6-nitro-7-sulfamoylbenzo[f]quinoxaline-2,3-dione (NBQX, Abcam), an AMPA receptor antagonist, in 0 mM [Mg$^{2+}$]/3 mM [Ca$^{2+}$] bath solution at −70 mV holding potential. Isolated AMPA and NMDA EPSCs were recorded from the same neurons sequentially by first applying D-AP5 then completely replacing it with NBQX. The NMDA/AMPA ratio for every neuron was calculated from the average amplitudes of the last 10 NMDA and AMPA events during 5 min D-AP5 or subsequent NBQX application. No postsynaptic currents were detected if D-AP5 and NBQX were applied together. All signals were digitized at 20 kHz, filtered at 2 kHz, and analyzed offline in Clampfit (Molecular Devices).

**Live cell fluorescence imaging and image analysis**. Live cell imaging was performed with 13–17 DIV cells using an Olympus BX-51WI microscope equipped with a 60× LUMPlanFl water-immersion objective (N.A. 0.9), a Sutter Instrument MP-78 xyz motorized stage, a Solamere laser combiner and launcher (405, 480, 561, and 640 nm lasers), a Yokogawa CSU-X1 spinning disk confocal head, and an Evolve 512 EMCCD (Photometrics). For TFC, GP, and Ca$^{2+}$ imaging, cells growing on bare or graphene-covered glass coverslips were pre-incubated with TFC (20 min, 1 µM, Avanti), C-laurdan (1 h, 1 µM, TP Probes), or Rhod-2 AM ester (30 min, 1 µM, Biotium) at 37 °C with 5% CO$_2$. After dye loading, coverslips were washed and mounted in an RC-26G imaging chamber (Warner Instruments) bottom-sealed with a 24 × 40 mm$^2$ size 0 cover glass (Fisher Scientific). The chamber was fixed in a PH-1 platform (Warner Instruments) fixed on the MP-78 stage and bath solutions were applied via gravity perfusion with a constant rate of ~50 µL/s. All perfusion lines were merged into an SHM-6 in-line solution heater (Warner Instruments). The temperatures of both the imaging chamber and the perfusion solution were maintained at 34 °C by a temperature controller (TC344B, Warner Instruments). For FM dye or Quantum dot (Qdot) loading of the evoked pool of synaptic vesicles, mounted coverslips were incubated with 10 µM FM1-43, 10 µM FM4-64, or 100 or 0.8 nM Qdots (Qdot 605, Thermo Fisher) for 2 min in high K$^+$ bath solution containing (in mM): 64 NaCl, 90 KCl, 2 MgCl$_2$, 2 CaCl$_2$, 10 N-2 hydroxyethyl piperazine-n-2 ethanesulphonic acid (HEPES), 10 glucose, 1 µM TTX, pH 7.35. For Qdot loading of the spontaneous pool of synaptic vesicles, cells were incubated with 100 nM Qdots for 15 min in normal Tyrode's solution. The Qdots used here had a hydrodynamic diameter of ~15 nm, limiting loading to one per synaptic vesicle (~25 nm luminal diameter)[58,59]. After FM or Qdot loading, cells were washed with normal Tyrode's solution containing 10 µM NBQX and 20 µM D-AP5 for 5 or 10 min, respectively, to remove surface dye/Qdots. Electric field stimulation (10 Hz, 70 V) was triggered by a 5-V 2-ms TTL pulse generated by the Clampex software 30 s after imaging began

and delivered via a pair of platinum wires attached to both sides of the imaging chamber by a Grass SD9 stimulator. Synchronization of perfusion with image acquisition was via a VC-6 valve system (Warner Instruments) and controlled in Clampex.

For TFC imaging, a 100 mW 480-nm laser (20% power) and a filter combination of DiC 500LX and Em 520/20 were used. Exposure time was 100 ms and the EM gain was 250 for all images. For every FOV, 10 repeated images were taken and averaged. For FM1-43 imaging, laser and filter sets were the same as those for TFC. Exposure time was 50 ms, the EM gain was 300, and the acquisition rate was 1 Hz. For GP imaging using C-laurdan, a 405-nm laser (40% power) and a filter combination of a DiC 409LP and an Em 440/40 or 483/32 (for blue or green channels, respectively) were usually used. For TFC pretreated cells, we used an arc lamp (LUMen 200, Prior) and a D350x excitation filter (Chroma) to avoid spectral cross-excitation of TFC. The exposure time was 50 ms with an EM gain of 900 for all images. For every FOV and each fluorescence channel, 10 repeated images were taken and averaged. For $Ca^{2+}$ imaging, a 100 mW 561-nm laser (20% power) and a filter combination of DiC 580LPXR and Em 605/52 were used. The exposure time was 150 ms, the EM gain was 500 for all images, and the acquisition rate was 1 Hz. For Qdot imaging, a 480-nm laser (80% power) and a filter combination of DiC 510LX and Em 605/10 were used. The exposure time was 200 ms, the EM gain was 250 for all images, and the acquisition rate was 5 Hz. For FM4-64 imaging, a 50-mW 640-nm laser (30% power) and a filter combination of DiC 660LX and Em 710/50 were used. The exposure time was 50 ms, the EM gain was 500 for all images, and the acquisition rate was 1 Hz. For static images, 10-frame stacks were averaged. All images were taken with the same acquisition settings between sample groups (laser intensity, exposure time, and EM gain).

Image analyses were performed in ImageJ. Four rectangular ROIs were drawn in cell-free regions in every FOV and their intensities averaged. For fluorescence imaging of TFC, C-laurdan (blue channel), FM1-43 (first 10 frames), FM4-64 (first 10 frames), and Rhod-2 AM, we pooled all background ROIs regardless of treatment differences in order to calculate the mean and standard deviation of the background intensity. Again, a masked threshold approach was applied in ImageJ, and the mean intensity plus two standard deviations was used as the common threshold for all images or image stacks. For every FOV, ROIs were generated by particle analysis based on a binary threshold mask. For TFC loading, watershed segmentation was used to generate ROIs. For FM1-43 and FM4-64, watershed segmentation and particle size limits (0.3–3 μm) were applied in ImageJ to isolate ROIs for synaptic boutons (~1 μm). ΔFM4-64 is defined here as the difference of FM4-64 fluorescence intensity before and after four rounds of exhaustive stimulation.

For Rhod-2, FM1-43, and FM4-64 time-lapse data, the average intensity from four background ROIs was subtracted from the average intensity of each individual ROI in the same FOV. Normalization was performed using the average intensity of the first 10 frames. For C-laurdan images, ROIs generated from blue channel images were used to analyze both channels. GP value was calculated as $I_{GP} = (I_{blue} - G \times I_{green})/(I_{blue} + G \times I_{green})$, in which $G$ is the sensitivity correction factor between the two channels[74]. $G$ was empirically determined by imaging 1 μM C-laurdan diluted in dimethyl sulfoxide (DMSO) using the prescribed protocol. Given $GP_{DMSO} = 0.006$, the $G$ value of our imaging setup was calculated using the following formula: $G = (I_{blue} \times (1 - GP_{DMSO}))/(I_{green} \times (1 + GP_{DMSO}))$. For Qdot images, FM4-64-defined ROIs were applied and the mean Qdot photoluminescence intensity in each ROI was calculated. Quantal analysis for single Qdots was performed as described previously[58,59]. Briefly, maximum likelihood estimates were used to fit Qdot counts to a distribution of intensities. Qdot intensities were binned every 30 a.u. without background subtraction. The estimated threshold based on the mean background signal plus two standard deviations was near 3000 a.u. by which we set the cut-off for a single Qdot. To analyze the behavior of vesicles labeled by single Qdots, we selected ROIs having only one Qdot. Time-dependent Qdot photoluminescence traces were extracted with a five-frame moving window.

**Statistical analysis**. All experiments were carried out blindly and repeated in at least three different batches of cultures ($N \geq 3$). All imaging experiments were repeated with at least three randomly selected coverslips per batch with one randomly chosen FOV per coverslip. No statistical methods were used to pre-determine sample size. All values presented are mean ± S.E.M. All fluorescence intensity values are background corrected except in Figure S5a. For two-group comparison of average values, data were first assessed for normality using the Lillefors test. Unpaired two-tailed t-tests or the Wilcoxin rank-sum test were used for two-group comparison of average values. Cumulative distribution functions (FM dye and Qdot imaging) or histograms (GP imaging) were used for two-group comparison of pooled values. Cumulative distributions are an accepted measure to provide an overview of intensity distributions from synapses[75–77] and more clearly demonstrate the overall trend of the individual data points than an average measure. For GP imaging data, all individual pixels from each FOV were pooled from each treatment condition as the GP distributions for the generation of test statistics. Kolmogorov–Smirnov tests were used to compare distributions. For Sholl analysis datasets, statistics were calculated using two-way analysis of variance (ANOVA) with repeated measures followed by the Bonferroni multiple

comparisons test. $\omega^2$ values were calculated as effect-size metrics for relevant ANOVA data. A one-way ANOVA and the Tukey–Kramer method as post-hoc analysis was used for three or more groups. Fisher z-tests were used to compare correlation coefficients. Because the sample sizes of several reported values are large and thus may overestimate true significance[78], Cohen's d-statistic[79] was also reported for some datasets.

**Data availability**. The data supporting the findings in this study are available from the corresponding author upon reasonable request.

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

## Acknowledgements

We thank I. Kristaponyte for technical assistance with cell culture. We thank H.E. Hamm and K.P.M. Currie for comments and discussions. The graphene devices were fabricated and characterized at the Center for Nanophase Materials Sciences at the Oak Ridge National Laboratory. We thank all members of the Zhang and Xu laboratories for their support. This work is funded by the National Science Foundation (ECCS-1055852 and CBET-1067213 to Y.-Q.X., CBET-1264982 to Y.-Q.X. and Q.Z.) and the National Institutes of Health (DA025143, OD00876101, and NS094738 to Q.Z.).

## Author contributions

K.E.K. and Q.Z. conceptualized the project and designed the experiments with input from all authors. K.E.K. and Q.Z. performed all imaging and spectrofluorometry experiments. R.M.L. performed all electrophysiology experiments. T.H. and D.Y. prepared and characterized graphene and performed all Raman spectroscopy measurements supervised by Y.-Q.X. Q.Z. supervised all other experiments. K.E.K., Y.-Q.X., and Q.Z. wrote and edited the manuscript with input from all authors.

## Additional information

**Competing interests:** The authors declare no competing financial interests.

