## [Peer Review File · Nature Communications]

Reviewers' comments:

Reviewer #1 (Remarks to the Author):

Kitko et al. demonstrated the ability for graphene substrates to alter the distribution of cholesterol within plasma membranes and its effect on modulating cellular responses in hippocampal neurons and fibroblast cells. Interestingly, increases in membrane cholesterol affected neurotransmitter release in neurons and activated GPCR signaling in fibroblasts. This paper is well-written and its findings are relevant to the biomaterials community. However, a recent paper [Veliev et al. Biomaterials 2016] has performed experiments like those presented in the current manuscript, which reduced the novelty of this work. Additionally, some questions related to data presentation and analysis exist and further examination is necessary to provide evidence for the described mechanisms. For those reasons, it is recommended to not accept this paper in its current form.

1. A prior computational study demonstrated the interactions of cholesterol and graphene [Zhang et al. J. Phys. Chem. B 2016]. Figure 1 only provides experimental evidence for this phenomenon.
2. A prior study has described the CVD deposition of uncoated graphene onto glass slides for hippocampal neuron culture [Veliev et al. Biomaterials 2016]. The authors should include this paper in their reference list and discuss since this paper affects part of the novelty of their study. Specifically, p.3 lines 54-55 and p.4 lines 68-89 as the cited paper studied the effects of culturing cells on CVD graphene. Characterization of the graphene films as demonstrated in Figure 2 is highly similar to the cited work.
3. The authors consistently rely on cell staining to generate their conclusions. However, the quantification of images throughout the paper for average staining (especially Figures 3a2, 4a4, 5b, and 6b) is not clear as the provided images are not readily identifiable as different. If the cell density per image is different, then it is possible that the results may be altered. How is this controlled for? Further discussion of how the authors used ImageJ and Matlab to analyze images is necessary.
4. Figure 3c1 and 4a1, identification of TFC punta is not clear from the image. It would be helpful for the authors to add arrows or other identification markers to the images to aid in data interpretation.
5. The brightness and resolution of many images is low. Increasing this quality will make the images clearer. To identify puncta, increased magnification would also make the conclusions more obvious.
6. It would be helpful for the authors to describe their vision for the translation of their findings towards the development of novel technologies. The impact of the findings is not clear and further discussion of the proposed uses would improve the discussion.
7. There is a typo on P.8 L 173.

Reviewer #2 (Remarks to the Author):

The submitted manuscript touches the important issue of interaction between graphene and eukaryotic cells. In view of fast development in production, commercial offers, near future proliferation and application of graphene in many areas of technology, the contact of humans with graphene included devices is almost inevitable. An increasing number of research work is conducted in the field of application of graphene in biotechnology and nanomedicine. In this context, the knowledge of influence of graphene sheets on living cells, especially on cell membrane is very desirable and necessary.

This manuscript presents the experimental approach to the particular problem of influence of

graphene on cholesterol embedded in biomembrane. So far, this problem was mainly studied using computer simulation methods (references [21] and [22] in the manuscript).

The Authors of submitted manuscript proved that graphene flakes are able to adsorb and extract cholesterol and its analogues from bio-environment. They also show that close contact of graphene surface with biomembrane leads to increased concentration of cholesterol in the membrane. The increased concentration of cholesterol in biomembrane affects several cell functions, including neurotransmitter release in neuron cell and activation of P2Y receptors in fibroblasts, as reported in the manuscript. This fact, to my knowledge so far unreported, that graphene can be considered as a potential tool for the control or manipulation of some functions of the eukaryotic cell, deserves attention and is important point in favor of the publication of this manuscript in the prestigious scientific journal Nature Communications. It could be a positive inspiration for the future research on the application of nanostructures in molecular medicine (nanomedicine) and biotechnology.

The statistical analysis is acceptable (standard). The presented conclusions are justified by presented experimental data.

I found the misprint on page 8, line 173, there is "lebel" and should be "level".

I recommend publication of the submitted manuscript.

Reviewer #3 (Remarks to the Author):

The authors present a study on the effect of monolayer graphene on cell membrane cholesterol levels of cells grown on top of it. In general (but see detailed comments below) the work appears to be competently performed and the manuscript (mostly) reads well. I am less excited about the broader significance of the work. Possibly, publishing the work in a more specialized journal (e.g., ACS Nano), rather than a more general journal like Nature Communications, would be a better option.

Scientifically, my main major comment concerns the effect of biomolecule adsorption. For nanoscale objects, adsorption of biomolecules to the object's surface, the formation of a biomolecular "corona" in the parlance of the field, is known to be a major determinant for how the object interact with cells. This is certainly something expected also of graphene. This aspect is currently not well-discussed in the manuscript and affects the overall conclusions at several levels. First, it is not clear to me if this effect would be present for the applications the authors envisage (in their Introduction), and the relevance or not of their study to these applications may need to be revised. Second, their experiments are, as far as I understand, performed under such circumstances where I expect the graphene surface to be covered with biomolecules, at least prior to the cells adherence. Their discussion about intramolecular interactions between graphene and cholesterol ("its planar tetracyclic rings structurally allow cholesterol") may consequently have to be revised. The same goes for their differentiation between their results and "protein-coated graphene" and guidance from previous simulation studies (Introduction and Conclusions). In general, I would advise the authors to reconsider these, and all other relevant, interpretations with this in mind.

Detailed comments:

There are several graphs throughout the manuscript where the y axis does not start from 0 (thus exaggerating small differences). I cannot find any justification for this, so I would suggest to simply let them start at 0.

"confirming direct contact between the graphene films and the cells": It is not clear to me why it implies direct contact, rather than just an effect.

Does C-laurdan distribute evenly? It would appear to me that this is important for the interpretation of the results.

"whole-cell patch-clamp recordings": Is the presence of graphene (especially given its novel electrical properties) as a substrate not a complication in such experiments?

Fig. 1a: Does the assay measure amount or concentration? This is important to interpret the highest fluorescence. Did I miss the Methods part for these experiments?

There are several graphs (e.g., Fig 1b, Fig. 3a2) which report "Adjusted" fluorescences. What does this imply?

Several results have indicated "n"s. It should be clarified what this refers to (number of cells? number of experiments?). There is some guidance in the Methods section, but it is still not unambiguous to me.

For the (quite substantial amount of) quantitative imaging data, more detail on ensuring that fluorescence levels can be directly compared is needed. The authors do suggest that imaging conditions were kept equal, but experience shows that this is not always enough. To give but one example, shifting the axial focus plane somewhat can give rather different fluorescence. There is a need of tightening the arguments surrounding these experiments, most likely with relevant controls etc.

Fig. 1b: At what time were these spectra recorded?

Fig. 3a2: Averaged in what way? If "n" refers to the number of cells, then my feeling is that that would be far too few to get meaningful results; the authors need to show that it is sufficient.

Fig. 3b1: What is the distribution over? Pixels? Please clarify also in all other cases (e.g., Fig. 4c1 and 4d1).

"Sholl analysis" and cluster analysis in Figure 4: Both needs to be introduced/explained, if only briefly, in the main text.

Fig. 4c1: I guess it is the cumulative distribution function. More importantly, why has the variable on the x axis been normalised? I do not see the scientific argument behind that.

Fig. 6d: I cannot see the results in this graph.

Fig. S2: Why does this graph start so much later than those in Fig. S2d2 or 4a2? Furthermore, it seems clear that the traces are different, contrary to how I understand the authors' interpretation.

Fig. S4: I think this argument needs refinement. For example, look at the fitted lines. The black one starts at unity (as it should, since the data has been normalised), but the red one does not. This leads to an exaggerated difference between the time constants evaluated from the fits. I think it would be more appropriate to use the original data, plotted in a semi-logarithmic plot, if the authors think evaluating the rate constants is really the best approach. In such a plot the differences should be apparent, even without fitting.

Fig. S5a: I find this (and the main figures that "depend" on it) rather confusing. First, it would appear the signal to background ratio is huge (388 vs 2,731). Second, where can I see those numbers in the actual graph? I see a main peak at around 3,100 and a second peak at around

3,700. Neither of these numbers agree with the authors'. This could be explained better.

Minor comments:

There are a significant number of acronyms in the manuscript which, especially in the latter parts, make it rather difficult to read. To give one example: "rate of FM4-64 loss, the overall FRF ratio, and the Pr_v [...] were all significantly reduced after MbCD". Given that the authors presumably want to reach readers from diverse backgrounds, I would suggest trying to increase the readability.

"then mixed with neuronal culture media": It should be clarified that the media contains cholesterol.

"Raman spectroscopy can be used to detect contact between biomolecules and graphene": I accept the experiments that follow, but this is too general a conclusion based on one biomolecule.

"we minimized the proliferation of astroglia...": I found this highly confusing at first and I think it needs to be clarified in the main text (not just the Methods section) that the authors are using a mixture of cells, not just neurons.

"astroglia-conditioned neuronal media": Please clarify what this means.

"C-laurdan fluorescence emission red-shifts as the lipid bilayer becomes less fluid": This could be clarified. Do the authors mean that C-laurdan red-shifts if it inserts into less fluid regions (where, presumably, cholesterol resides)?

"high cholesterol lebel": Typo.

Fig. 3c1: I suggest to switch the order between graphene and glass for consistency.

Fig. 3c4: Please state what the lines are in the figure caption.

Reviewers' comments:

Reviewer #1 (Remarks to the Author):

Kitko et al. demonstrated the ability for graphene substrates to alter the distribution of cholesterol within plasma membranes and its effect on modulating cellular responses in hippocampal neurons and fibroblast cells. Interestingly, increases in membrane cholesterol affected neurotransmitter release in neurons and activated GPCR signaling in fibroblasts. This paper is well-written and its findings are relevant to the biomaterials community. However, a recent paper [Veliev et al. Biomaterials 2016] has performed experiments like those presented in the current manuscript, which reduced the novelty of this work. Additionally, some questions related to data presentation and analysis exist and further examination is necessary to provide evidence for the described mechanisms. For those reasons, it is recommended to not accept this paper in its current form.

1. A prior computational study demonstrated the interactions of cholesterol and graphene [Zhang et al. J. Phys. Chem. B 2016]. Figure 1 only provides experimental evidence for this phenomenon.

As described in our introduction, this and other computational results inspired us to empirically test a predicted graphene-cholesterol interaction and to investigate if this interaction makes cholesterol a mediator of graphene's impact on a variety of cell functions.

2. A prior study has described the CVD deposition of uncoated graphene onto glass slides for hippocampal neuron culture [Veliev et al. Biomaterials 2016]. The authors should include this paper in their reference list and discuss since this paper affects part of the novelty of their study. Specifically, p.3 lines 54-55 and p.4 lines 68-89 as the cited paper studied the effects of culturing cells on CVD graphene. Characterization of the graphene films as demonstrated in Figure 2 is highly similar to the cited work.

We apologize for not referencing this paper, which first reported primary neuronal culture on pristine CVD graphene. We have now cited and discussed it in our revised manuscript. We do believe that the focus of our research and our major findings are very different from that study:

- (1) As indicated in the title ("Impact of crystalline quality on neuronal affinity of pristine graphene"), Veliev et al. demonstrated that "the crystallinity of CVD grown graphene plays an important role in neuronal attachment, outgrowth and axonal specification". Our study is thematically different: to understand the cellular mechanisms underlying graphene's impact and through which biomolecules these effects are mediated.
- (2) Although we have examined neuronal development, the vast majority of our study was done using mature and synaptically connected neurons, which are necessary for our focus on synaptic transmission.
- (3) In addition to neurons, we used fibroblast cells to investigate P2YR-mediated Ca^{2+} -signaling. This set of results suggests complex effects of graphene depending upon the roles of cholesterol within the cell membrane.

Thus, we believe that the overlap between Veliev's paper and ours is largely limited to the graphene-cell configuration (i.e. primary neuronal cultures on uncoated CVD graphene) and graphene characterization.

3. The authors consistently rely on cell staining to generate their conclusions. However, the quantification of images throughout the paper for average staining (especially Figures 3a2, 4a4, 5b, and 6b) is not clear as the provided images are not readily identifiable as different. If the cell density per image is different, then it is possible that the results may be altered. How is this controlled for? Further discussion of how the authors used ImageJ and Matlab to analyze images is necessary.

We apologize for the low image quality in our initial submission. We have now uploaded the figures individually. In the revision, we have included enlarged and higher-resolution sample images for better clarity. In the new methods section, we have expanded the image acquisition and analysis sections. We took a series of measures to ensure a fair comparison and reliable conclusions:

- (1) All experiments were conducted using at least 3 batches of cell culture and 2 coverslips for each group in every batch.
- (2) Except for Figure 4a4 (immunostaining), all fields of view (FOVs) were randomly selected. For cells in 4a4, we selected biocytin-filled neurons with the somas in the center of the FOVs.
- (3) During image acquisition, all parameters, including excitation light intensity, fluorescence filters, exposure time, and detector gain were kept the same in each dataset.
- (4) For each FOV, at least three background regions of interest (ROIs) from cell-free areas were randomly selected to calculate the average background intensity, which was subtracted from the mean intensities of individual ROIs.
- (5) All results shown in Fig. 3a2, 4a4, 5b and 6b were consistent with results from other independent datasets, including GP imaging, FM&Qdot imaging and electrophysiology.

We did not control for cell density intentionally. However, we have quantified the number of

cells in every FOV for the image sets used in Fig. 3a2, 4a4, 5b and 6b. As shown in **Figure a-d**, cell numbers between different groups were statistically similar ($p_a = 0.675$, Wilcoxin rank-sum test; $p_b = 0.88$, Wilcoxin rank-sum test; $p_c > 0.10$, one-way ANOVA; $p_d > 0.10$, one-way ANOVA).

4. Figure 3c1 and 4a1, identification of TFC punta is not clear from the image. It would be helpful for the authors to add arrows or other identification markers to the images to aid in data interpretation.

Thanks for the suggestion. We have added identification markers in those images accordingly.

5. The brightness and resolution of many images is low. Increasing this quality will make the images clearer. To identify puncta, increased magnification would also make the conclusions more obvious.

We have changed or adjusted the images and added zoom-ins to improve clarity. To be noted, such adjustment does not affect our image analyses, which are completely based on original fluorescence intensities.

6. It would be helpful for the authors to describe their vision for the translation of their findings towards the development of novel technologies. The impact of the findings is not clear and further discussion of the proposed uses would improve the discussion.

We have added a few sentences in the discussion about potential graphene applications based on our findings.

7. There is a typo on P.8 L 173.

Corrected.

Reviewer #2 (Remarks to the Author):

The submitted manuscript touches the important issue of interaction between graphene and eukaryotic cells. In view of fast development in production, commercial offers, near future proliferation and application of graphene in many areas of technology, the contact of humans with graphene included devices is almost inevitable. An increasing number of research work is conducted in the field of application of graphene in biotechnology and nanomedicine. In this context, the knowledge of influence of graphene sheets on living cells, especially on cell membrane is very desirable and necessary.

This manuscript presents the experimental approach to the particular problem of influence of graphene on cholesterol embedded in biomembrane. So far, this problem was mainly studied using computer simulation methods (references [21] and [22] in the manuscript).

The Authors of submitted manuscript proved that graphene flakes are able to adsorb and extract cholesterol and its analogues from bio-environment. They also show that close contact of graphene surface with biomembrane leads to increased concentration of cholesterol in the membrane. The increased concentration of cholesterol in biomembrane affects several cell functions, including neurotransmitter release in neuron cell and activation of P2Y receptors in fibroblasts, as reported in the manuscript. This fact, to my knowledge so far unreported, that graphene can be considered as a potential tool for the control or manipulation of some functions of the eukaryotic cell, deserves attention and is important point in favor of the publication of this manuscript in the prestigious scientific journal Nature Communications. It could be a positive inspiration for the future research on the application of nanostructures in molecular medicine (nanomedicine) and biotechnology.

The statistical analysis is acceptable (standard). The presented conclusions are justified by presented experimental data.

I found the misprint on page 8, line 173, there is "lebel" and should be "level".
We have corrected this typo in the revised text.

I recommend publication of the submitted manuscript.

Reviewer #3 (Remarks to the Author):

The authors present a study on the effect of monolayer graphene on cell membrane cholesterol levels of cells grown on top of it. In general (but see detailed comments below) the work appears to be competently performed and the manuscript (mostly) reads well. I am less excited about the broader significance of the work. Possibly, publishing the work in a more specialized journal (e.g., ACS Nano), rather than a more general journal like Nature Communications, would be a better option.

Scientifically, my main major comment concerns the effect of biomolecule adsorption. For nanoscale objects, adsorption of biomolecules to the object's surface, the formation of a biomolecular "corona" in the parlance of the field, is known to be a major determinant for how the object interact with cells. This is certainly something expected also of graphene. This aspect is currently not well-discussed in the manuscript and affects the overall conclusions at several levels. **First**, it is not clear to me if this effect would be present for the applications the authors envisage (in their Introduction), and the relevance or not of their study to these applications may need to be revised. **Second**, their experiments are, as far as I understand, performed under such circumstances where I expect the graphene surface to be covered with biomolecules, at least prior to the cells adherence. Their discussion about intramolecular interactions between graphene and cholesterol ("its planartetracyclic rings structurally allow cholesterol") may consequently have to be revised. The same goes for their differentiation between their results and "protein-coated graphene" and guidance from previous simulation studies (Introduction and Conclusions). In general, I would advise the authors to reconsider these, and all other relevant, interpretations with this in mind.

Following the reviewer's suggestion, we have included the issues of graphene-cell contact, biomolecular "corona", and relevance to graphene applications in the revised manuscript. Below are our answers to his/her specific concerns.

First, it is possible that biomolecules can accumulate on the graphene surface in some applications, especially those involving long-term contact with biological samples. Whether accumulated biomolecules can completely block the access to graphene surface depends on many factors like biomolecules' association/dissociation rates and incubation time. For graphene electrodes, the fact that it was possible to detect membrane potential changes *in vitro* and *in vivo*^{1,2} strongly suggested that the graphene surface remains accessible after long-term contact with cells or tissue. In our study, we mimicked these scenarios by plating cells directly on bare pristine graphene films over days of culture, which allowed the formation of a biomolecular corona. Our findings suggested that a biomolecule corona does not necessarily prevent contact between the cell surface and graphene. Graphene did change the cell membrane, especially its cholesterol content, which caused a variety of cellular changes. For graphene-based applications, our results will help to highlight the role of the cell membrane in mediating the effects of graphene, to achieve better interpretations of cellular changes caused by graphene-based devices, and to suggest new usages for graphene.

Second, results from our and others' studies suggest that cells still have access to and are affected by graphene surfaces during days of culture. **(1)**, dissociated hippocampal cells were plated on bare graphene or bare glass coverslips that had never been pre-exposed to any biological materials. Thus, there was not a pre-adsorbed coating before cell deposition. **(2)**, Hu et al showed that the time for protein corona formation on graphene oxide (more hydrophilic, thus more adsorptive to proteins than graphene) mixed with serum-containing media is about 30 minutes³. We have tested cell attachment to bare glass (a less favorable surface for attachment than graphene, as demonstrated in Veliev et al.) and found a similar time course for neuronal adhesion (**Figure e**). Therefore, the rate of cell adhesion to bare graphene is at least similar to the formation of a protein corona. **(3)**, Veliev et al⁴ recently showed that the surface crystallinity of graphene played a major role neuronal adhesion and neurite outgrowth in cell culture, which took days to grow. Since they also plated cells directly on bare graphene, it is clear that the formation of any biomolecular corona during days of culture was not enough to attenuate the impact of graphene surface crystallinity. Of note, our research focus and major findings are significantly different from that report (please see reply to reviewer #1). There are also biosensor applications which still show great sensitivity in spite of any protein corona formation^{1,2}. **(4)**, our Raman spectrum data (**Fig. 2**) demonstrated that, even after days of culture, the cell-free areas of graphene were very similar to the Matrigel-free areas that were never exposed to serum-containing media. This indicates that the biomolecular corona is at least less dense than a Matrigel coating. We have incorporated the above discussion into the revised manuscript.

Detailed comments:

There are several graphs throughout the manuscript where the y axis does not start from 0 (thus exaggerating small differences). I cannot find any justification for this, so I would suggest to simply let them start at 0. We have revised the graphs as the reviewer suggested.

“confirming direct contact between the graphene films and the cells”: It is not clear to me why it implies direct contact, rather than just an effect. We have revised that statement.

Does C-laurdan distribute evenly? It would appear to me that this is important for the interpretation of the results.

Previous studies have demonstrated that Laurdan molecules distribute homogeneously in the lipid bilayer and move freely, color changes come as a result of a molecule diffusing into an area of increased lipid packing, whereupon a dipole change occurs in turn causing a spectral shift⁵. We find the diffuse staining in the blue channel (below) to be in good agreement. **Figure f** demonstrates that: (1) C-Laurdan distributes relatively evenly across the cell membrane as seen

in the green fluorescence channel, (2) its blue fluorescence is relatively less even, and (3) the brighter areas in the blue channel largely overlap with FM4-64 labelled synapses, which is expected as synaptic membranes have a higher concentration of cholesterol^{6,7}.

“whole-cell patch-clamp recordings”: Is the presence of graphene (especially given its novel electrical properties) as a substrate not a complication in such experiments?

Graphene was underneath neurons, and the recording electrodes approached neurons from above. For whole-cell patch-clamp recording, electrodes seal onto the apical plasma membrane (without cell penetration). Thus, there was no direct contact between graphene and the electrodes which means graphene cannot directly interfere with recordings. Also, it is unlikely that graphene in contact with neuronal membrane affects the propagation of neuronal electrical signals, because we observed no difference in the amplitude of sEPSCs, which were initiated at the dendritic terminals and propagated to the neuronal soma where the recording electrodes resided.

Fig. 1a: Does the assay measure amount or concentration? This is important to interpret the highest fluorescence. Did I miss the Methods part for these experiments?

We used a commercially available assay kit (ThermoFisher Scientific, <https://www.thermofisher.com/order/catalog/product/A12216>) that measures cholesterol concentration in the sample⁸. We have included a detailed description of this assay in the revised methods section. A serial dilution of cholesterol standard was used to generate a calibration curve (open gray circles and gray line representing linear regression fitting in Fig. 1a). Fluorescence emission was measured and converted to a concentration based on the calibration curve.

There are several graphs (e.g., Fig 1b, Fig. 3a2) which report “Adjusted” fluorescences. What does this imply?

In Fig. 1b, “adjusted” refers to TFC fluorescence being corrected to account for the difference in

absorbance between different concentrations of graphene solution (Fig. S1a). This is also applicable for Fig. S1e&f. In Fig. 3a2 and S2b1&d1, “adjusted” meant background subtraction. Since background subtraction was applied to all of our image analyses, we have removed this term for consistency and clarified our methods’ sections accordingly.

Several results have indicated “n”’s. It should be clarified what this refers to (number of cells? number of experiments?). There is some guidance in the Methods section, but it is still not unambiguous to me.

In the revised manuscript, we have annotated “n”’s in the text and/or figure legends.

For the (quite substantial amount of) quantitative imaging data, more detail on ensuring that fluorescence levels can be directly compared is needed. The authors do suggest that imaging conditions were kept equal, but experience shows that this is not always enough. To give but one example, shifting the axial focus plane somewhat can give rather different fluorescence. There is a need of tightening the arguments surrounding these experiments, most likely with relevant controls etc.

Results involving the direct comparison of fluorescence levels are Fig. 3a2&c2&c4, 4a4, 5b, 6b, S2b1&d1. As we described in our reply to reviewer #1 (comment 3), we used the same settings for imaging and analysis for every dataset, included both biological and technical replicates, chose FOVs or cells of interest randomly, automatically select ROIs with the same criteria (e.g. intensity threshold) among different treatments, and adjusted all ROI fluorescence intensities in every FOV by subtracting background intensities obtained in the same FOV. We have greatly expanded our methods section to describe this in more detail.

Figure g. Background fluorescence is sensitive to focus change. (a) Background ROIs were cell-free areas. (b) Inset images. (c) Background signal changes as a function of focal plane change. (d) Syp immunostaining fluorescence is also sensitive to focal plane.

the case based on our quantification (**Figure h**).

It is unlikely that there was a systematic bias in axial focus plane between the glass and graphene groups, as our only criteria for choosing focal plane was to get the largest amount of the sample area to approximately the same sharpness. And CVD graphene is too thin to alter sample thickness. When checking background fluorescence intensities, we noticed that they were correlated to axial focus plane (**Figure g**). We would expect a significant difference in background fluorescence intensities between the groups if there had been a systematic focus plane difference. However, this is not

Fig. 1b: At what time were these spectra recorded?

The spectra were recorded after 1-hour incubation. This information is now included in the figure legend and main text.

Fig. 3a2: Averaged in what way? If “n” refers to the number of cells, then my feeling is that that would be far too few to get meaningful results; the authors need to show that it is sufficient.

We apologize for the confusion. “n” in the original figure legend referred to the number of FOVs. To be consistent and clear, we now use “n” as the total number of cells or ROIs analyzed, $n_{\text{glass}} = 96$ and $n_{\text{graphene}} = 111$ (cells) for this figure panel. In the updated figure, we use a beeswarm plot for improved transparency.

Fig. 3b1: What is the distribution over? Pixels? Please clarify also in all other cases (e.g., Fig. 4c1 and 4d1).

For Fig. 3b1 and b2, the histogram of GP values is over all pixels in the FOV, same for Fig. 5c and 6c. Fig. 4c1 are the cumulative distributions of normalized (see below) FM1-43 fluorescence intensity. Fig. 4d1 is the cumulative distributions of Qdot photoluminescence intensity in synaptic areas defined by FM4-64 staining. Fig. S5c are the cumulative distributions of the times of individual Qdots loss counted from the beginning of the field stimulation.

“Sholl analysis” and cluster analysis in Figure 4: Both needs to be introduced/explained, if only briefly, in the main text.

We have added sections in both the main text and methods section to explain both Sholl and cluster analysis.

Fig. 4c1: I guess it is the cumulative distribution function. More importantly, why has the variable on the x axis been normalised? I do not see the scientific argument behind that.

Yes, it is the cumulative distribution function over FM1-43 defined ROIs. We normalized the value on the x-axis to be consistent with Fig.

4c2. The normalization is that the different FM1-43 intensities between the pre-stimulation baseline (defined as 1) and the post-stimulation (after three rounds of 90 mM K^+ stimulation) baseline (defined as 0) corresponding to FM1-43 residing in all releasable synaptic vesicles in every synaptic bouton. This normalization is often used to control for the variability in FM1-43 loading (e.g. dye concentration and stimulation strength)⁹. In our experiments, this normalization does not affect the overall outcome, as shown by the plot of absolute FM1-43 fluorescence values (**Figure i**).

Figure i. Cumulative distributions of FM1-43 intensities in glass (black) and graphene (red) group.

Fig. 6d: I cannot see the results in this graph.

We have revised the graph with different color designations to improve readability.

Fig. S2: Why does this graph start so much later than those in Fig. S2d2 or 4a2? Furthermore, it seems clear that the traces are different, contrary to how I understand the authors' interpretation.

We have replotted figures 4a2, S2b2 and S2d2 so that they have the same axis scales. The traces do differ in the sense that arborization becomes greater for more mature neurons. 4a2 is 13-17 DIV neurons, which were matured and synaptically connected, and S2b2 and S2d2 are for 3 and 7 DIV respectively, when neurons were developing. Only in the revised S2d2 (7DIV), there was a statistically significant difference at 48.5 μm away from the soma, which does not change our conclusion that, in our culture preparation, graphene does not change overall neuronal morphology. This is now included in the figure legend for Figure S2.

Fig. S4: I think this argument needs refinement. For example, look at the fitted lines. The black one starts at unity (as it should, since the data has been normalised), but the red one does not. This leads to an exaggerated difference between the time constants evaluated from the fits. I think it would be more appropriate to use the original data, plotted in a semi-logarithmic plot, if the authors think evaluating the rate constants is really the best approach. In such a plot the differences should be apparent, even without fitting.

We include a revised plot (**Figure j**) for the stimulation period, in an Arrhenius style using raw fluorescence values, based on the reviewer recommendations. Linear regression lines are also included.

Fig. S5a: I find this (and the main figures that “depend” on it) rather confusing. First, it would appear the signal to background ratio is huge (388 vs 2,731). Second, where can I see those numbers in the actual graph? I see a main peak at around 3,100 and a second peak at around 3,700. Neither of these numbers agree with the authors’. This could be explained better.

The data were Qdot photoluminescence intensity directly extracted from original images without background subtraction. However, the x-axis was mislabeled. It is now corrected.

Minor comments:

There are a significant number of acronyms in the manuscript which, especially in the latter parts, make it rather difficult to read. To give one example: “rate of FM4-64 loss, the overall FRF ratio, and the Pr_v [...] were all significantly reduced after MbCD”. Given that the authors presumably want to reach readers from diverse backgrounds, I would suggest trying to increase the readability.

Thanks for the suggestion. We have revised the main text to improve overall readability.

“then mixed with neuronal culture media”: It should be clarified that the media contains cholesterol.

We have clarified this in the main text and methods sections.

“Raman spectroscopy can be used to detect contact between biomolecules and graphene”: I accept the experiments that follow, but this is too general a conclusion based on one biomolecule.

We have revised that.

“we minimized the proliferation of astroglia...”: I found this highly confusing at first and I think it needs to be clarified in the main text (not just the Methods section) that the authors are using a mixture of cells, not just neurons.

Thanks for the suggestion. We have revised that.

“astroglia-conditioned neuronal media”: Please clarify what this means.

It is the same neuronal media but pre-incubated for 48-hours with sister cultures plated on Matrigel-coated coverslips without mitosis inhibitor treatment. These sister cultures are relatively enriched in astrocytes. This pre-incubated media contains many factors secreted from astrocytes that are supportive of growth. We have revised the main text and methods to clarify.

“C-laurdan fluorescence emission red-shifts as the lipid bilayer becomes less fluid”: This could be clarified. Do the authors mean that C-laurdan red-shifts if it inserts into less fluid regions (where, presumably, cholesterol resides)?

We apologize for the confusion. C-laurdan exhibits a red-shift when located in less ordered regions of the lipid bilayer¹⁰.

“high cholesterol lebel”: Typo.

Corrected.

Fig. 3c1: I suggest to switch the order between graphene and glass for consistency.

Thanks. We switched the order.

Fig. 3c4: Please state what the lines are in the figure caption.

The lines represent linear regression fittings. We added this with the parameters to the legend.

References

- 1 Kuzum, D. *et al.* Transparent and flexible low noise graphene electrodes for simultaneous electrophysiology and neuroimaging. *Nat Commun* **5**, 5259, doi:10.1038/ncomms6259 (2014).
- 2 Zhang, Y. *et al.* Probing electrical signals in the retina via graphene-integrated microfluidic platforms. *Nanoscale* **8**, 19043-19049, doi:10.1039/c6nr07290a (2016).
- 3 Hu, W. *et al.* Protein corona-mediated mitigation of cytotoxicity of graphene oxide. *ACS Nano* **5**, 3693-3700, doi:10.1021/nn200021j (2011).
- 4 Veliev, F., Briancon-Marjollet, A., Bouchiat, V. & Delacour, C. Impact of crystalline quality on neuronal affinity of pristine graphene. *Biomaterials* **86**, 33-41, doi:10.1016/j.biomaterials.2016.01.042 (2016).
- 5 Sanchez, S. A., Tricerri, M. A. & Gratton, E. Laurdan generalized polarization fluctuations measures membrane packing micro-heterogeneity in vivo. *Proc Natl Acad Sci U S A* **109**, 7314-7319, doi:10.1073/pnas.1118288109 (2012).
- 6 Takamori, S. *et al.* Molecular Anatomy of a Trafficking Organelle. *Cell* **127**, 831-846, doi:<http://dx.doi.org/10.1016/j.cell.2006.10.030> (2006).
- 7 Wilhelm, B. G. *et al.* Composition of isolated synaptic boutons reveals the amounts of vesicle trafficking proteins. *Science* **344**, 1023-1028, doi:10.1126/science.1252884 (2014).
- 8 Amundson, D. M. & Zhou, M. Fluorometric method for the enzymatic determination of cholesterol. *Journal of biochemical and biophysical methods* **38**, 43-52 (1999).
- 9 Gaffield, M. A. & Betz, W. J. Imaging synaptic vesicle exocytosis and endocytosis with FM dyes. *Nat. Protocols* **1**, 2916-2921, doi:http://www.nature.com/nprot/journal/v1/n6/supinfo/nprot.2006.476_S1.html (2007).
- 10 Kim, H. M. *et al.* A two-photon fluorescent probe for lipid raft imaging: C-laurdan. *Chembiochem* **8**, 553-559, doi:10.1002/cbic.200700003 (2007).

Reviewers' comments:

Reviewer #1 (Remarks to the Author):

The revised manuscript by Kitko et al. is significantly improved from its initial submission and the authors have adequately addressed my original concerns. I recommend publication of this manuscript in its current form.

Reviewer #2 (Remarks to the Author):

I suggest to publish the manuscript, taking into account the improvements made by authors.

Reviewer #3 (Remarks to the Author):

The authors have answered most of my original queries in their rebuttal (though some remain unclear or only partly made it into the actual manuscript; see below). Nevertheless, with the improved figures (e.g., proper axes and sharper images) I am now less convinced of the full argument. I think it is clear that there is an effect, but I feel that there are too many unconvincing and, possibly, contradictory results to justify publication at this time.

For example, while Figure 3b shows a statistically significant difference, it is clearly not a substantial difference. This result alone would suggest that while it may be the case that "graphene increased neuronal cholesterol", it is certainly not a big effect. It also appears to be inconsistent with Figure 8b (see below). Several other results, while, again, statistically significant, are also not substantial (Figure 6e and, notably, Figure S2c and f which present really large samples and hence small p values, but very small differences). In contrast, Figure 7-9 show much more larger effects.

More detailed comments:

I am mostly satisfied by the authors' reply to my comment about potential corona formation; however, their statement (p. 16): "Hu et al. recently showed that it took about 30 minutes for a stable protein corona to form [on] graphene oxide" I find rather surprising. I would guess the authors are referring to Figure 1c in that paper, but surely that figure suggests 10 min. I would also note that that is for complete saturation and the amount that remains after centrifugation (that is, very strongly bound protein).

Figure 4a: I cannot see any TFC puncta. There are, of course, FM4-64 puncta and TFC fluorescence overlaps with those, but I cannot find any TFC puncta themselves. Consequently, I have difficulty interpreting also, e.g., Figure 4c which is based upon TFC puncta.

Figure 4d: It would appear something is wrong with the lines given in the figure caption. That is, the lines given do not reproduce the lines shown in the graph, or something is left unspecified (e.g., what is the delta on the x axis?). As currently written, I would say the ratio is higher for glass (1937.9 is higher than 1752.8), which is not what I think the authors are saying.

Figure 6d: I am confused by this result. It appears to me that the authors have already presented results on synaptic vesicle density in Figure 4a (see my earlier comment where I argued that what the authors refer to as TFC puncta actually are FM4-64 puncta and hence synaptic vesicles). Those results showed a significant difference between glass and graphene, contrary to this results. It would be useful to the reader to understand the difference between these two sets of results.

p. 11 "confirmed single-Qdot labeling (Figure S5a) at this concentration": I found this interpretation highly surprising. Surely there is a large number of doublets?

Figure 8b: As far as I understand, these results (partly) repeat those shown in Figure 3b. However, in Figure 3b the results for filippin staining on graphene and glass are largely similar (as already noted), while in Figure 8b they are substantially different. It would be helpful if the authors could elaborate on the difference.

Minor comments

p. 5: Please clarify that neuronal culture media contains cholesterol; it makes the experiment easier to understand.

p. 6-7 "which is consistent with the formation of a protein corona on the graphene surface": I understand that this was inserted to satisfy my comment, but I do not understand what it is supposed to mean.

p. 7 first paragraph: Please rewrite for clarity vis-a-vis neurons, glial cells and hippocampal cultures.

Figure 3 caption: What does the double asterisks mean? In the case of panel d, what is it that is statistically different? The mean GP?

Figure S3: Is the figure not mislabelled with respect to DAP5 and NBQX?

Figure 7a2, b2 and d2 and Figure S5d: It must be noted what red and black represent. I cannot find this in the captions.

Figure S4: I find the linear fits presented meaningless; clearly the data is not linear over the whole time duration. The authors should restrict the range to the latter part of the curves, where the data in fact is linear. Alternatively, simply remove the fits.

Figure S5a: The description is still not clear. Based on the authors' answer I understand what they mean, but the figure caption is not clear: "Quantal analysis [...] indicates that the mean photoluminescence intensity of loaded single Qdots was 378 ± 41 a.u.". In the graph the peak is around 3100. I would strongly urge the authors to go through the whole manuscript and sort out how they describe fluorescence intensities, background subtracted or not.

p. 11: Reads "Figure 4d2"; should probably read "Figure 7d2".

p. 12: Reads "S8b&c"; should probably read "S5b&c".

Figure S5c: What is the grey line? I cannot find this in the caption.

p. 17 "Although graphene has been demonstrated to damage bacterial cell membranes it has been shown to have few adverse effects on eukaryotic cells [...]. This may be explained in part by [...] the fact that cholesterol is abundant in eukaryotic cell membranes but absent in most prokaryotic membranes.": I do not understand what the authors are suggesting. If there is no cholesterol graphene causes damage to cells?

p. 18 "this study": Meaning reference 21?

Reviewers' comments:

Reviewer #1 (Remarks to the Author):

The revised manuscript by Kitko et al. is significantly improved from its initial submission and the authors have adequately addressed my original concerns. I recommend publication of this manuscript in its current form.

We thank the reviewer.

Reviewer #2 (Remarks to the Author):

I suggest to publish the manuscript, taking into account the improvements made by authors.

We thank the reviewer.

Reviewer #3 (Remarks to the Author):

The authors have answered most of my original queries in their rebuttal (though some remain unclear or only partly made it into the actual manuscript; see below). Nevertheless, with the improved figures (e.g., proper axes and sharper images) I am now less convinced of the full argument. I think it is clear that there is an effect, but I feel that there are too many unconvincing and, possibly, contradictory results to justify publication at this time.

For example, while Figure 3b shows a statistically significant difference, it is clearly not a substantial difference. This result alone would suggest that while it may be the case that “graphene increased neuronal cholesterol”, it is certainly not a big effect. It also appears to be inconsistent with Figure 8b (see below). Several other results, while, again, statistically significant, are also not substantial (Figure 6e and, notably, Figure S2c and f which present really large samples and hence small p values, but very small differences). In contrast, Figure 7-9 show much more larger effects.

We highly appreciate the referee's methodical review and insightful comments.

For all listed statistics, we did not predetermine sample sizes. In every FOV, the number of cells or synaptic boutons was random. We have now included a statement about sampling in the methods section. For every related test, we used at least three different batches of cell culture, at least three randomly picked coverslips from every batch, and three randomly chosen fields of view (FOVs) in every coverslip for imaging. In every FOV, the number of cells or synapses was random. Since there were often hundreds of Syp puncta in every FOVs, we ended up with large sample sizes in Fig. 6e, S2c and S2f.

As a general rule, we take into consideration sample sizes from similar studies that use similar methods¹. Reports using Filipin staining to detect changes in cholesterol content often had similar sample sizes. In a recent Cell paper, Chu et al imaged at least 30 cells for every Filipin staining experiment and did at least 3 independent experiments in their study about membrane cholesterol transportation². Similar sample sizes have also been used in studies about the effects of graphene and graphene oxide on cultured cells. For

instance, Veliev et al. quantified > 140 neurons per condition for measuring neuronal density³. And in a related report about graphene oxide's effect on neurons, Rauti et al. quantified 13 FOVs/condition for immunofluorescence labeling (exact number of cells not provided⁴), which is well aligned with the sampling method we used.

The difference in each set of experiments are statistically significant, and the unpaired two-tailed t-test we used do take into account sample size (i.e. degrees of freedom). For further transparency, we include a table of mean \pm MSE, n, and *p* values for listed figures.

Fig.	n _{glass}	Mean _{glass}	n _{graphene}	Mean _{graphene}	p
3b	96	129.8 \pm 1.3	111	139.5 \pm 2.5	0.0012
6e	11051	2350 \pm 14	10773	2400 \pm 20	9.1e-14
S2c	1524	1350 \pm 15	1715	1400 \pm 13	9.2e-28
S2f	4381	5100 \pm 23	3990	5300 \pm 33	2.4e-16
8b	188	45.97 \pm 1.98	107	105.47 \pm 3.43	1.0e-20

Figure R1. Filipin staining at somas and neurites (based on the original image sets used for Fig. 8b). The neurite data is adopted from Fig. 8b. **, *p* < 0.01; ***, *p* < 0.005, two-tailed *t*-test.

There may be an explanation for the differences mentioned by the reviewer. For Fig. 3b, we measured Filipin staining at neuronal somas that were readily identifiable; whereas for Fig. 8b, we measured Filipin staining at neurites as we focused on presynaptic effects. Notably, somas contain a large number of membrane-bound organelles, whose cholesterol is also labeled by Filipin, which might explain the higher overall Filipin intensity and smaller difference (Fig. 3b). To test that, we measured Filipin labeling at neuronal somas in the Fig. 8b image sets, which show a similar trend (Fig. R1).

Fig. 7 is FM1-43 or Qdot loading, which cannot be directly compared with Filipin or Synaptophysin results taken from the fixed cells. Fig. 9b was done using 3T3 cells, whose membrane cholesterol concentration and distribution are different from those of neurons.

More detailed comments:

I am mostly satisfied by the authors' reply to my comment about potential corona formation; however, their statement (p. 16): "Hu et al. recently showed that it took about 30 minutes for a stable protein corona to form [on] graphene oxide" I find rather surprising. I would guess the authors are referring to Figure 1c in that paper, but surely that figure suggests 10 min. I would also note that that is for complete saturation and the amount that remains after centrifugation (that is, very strongly bound protein).

The "30 minutes" was cited from Hu et al's report⁵: "Additionally, GO-FBS protein binding reached equilibrium within 30 min (Fig.1c)." (line 9, page 3694).

We also note that in our Fig. S6, 10 minutes was the time for cell attachment to graphene to approximately reach a steady state. Both figures are included below for comparison.

Figure S6. Dissociated hippocampal neurons were plated on bare glass coverslips and incubated in plating media for the designated periods of time before washing with Hank's solution. Cells in randomly chosen FOVs were counted to calculate cell densities.

Figure 1c from Hu et al (2011). “(c) FBS protein loading ratio on the surfaces of GO at different incubation times. The GO was incubated with FBS proteins at 37 °C for 1, 5, 10, 30, 60, and 120 min. Then the amount of FBS proteins in the supernatant was determined *via* the Bradford method after centrifugation.”

Figure 4a: I cannot see any TFC puncta. There are, of course, FM4-64 puncta and TFC fluorescence overlaps with those, but I cannot find any TFC puncta themselves. Consequently, I have difficulty interpreting also, e.g., Figure 4c which is based upon TFC puncta.

It is true that the TFC staining was not as discrete as FM4-64, because the latter was mostly trapped in synaptic vesicles. But it is clear that there were areas with stronger TFC labeling (middle column in Fig. 4a) along neurites, which we now call “TFC-rich areas”. The TFC fluorescence intensities shown in Fig. 4b&d were measured in the ROIs defined by FM4-64 puncta (i.e. synaptic boutons). To analyze TFC-rich areas (Fig. 4c), we performed similar selection procedures for FM4-64 and TFC (Fig. R2), i.e. applying the same intensity threshold to all images taken from both groups and selecting ROIs in ImageJ. We have updated our figures and text accordingly.

Figure R2. TFC-rich area selection. **a**, original image. **b**, binary image after thresholding and applying watershed. **c**, selection of TFC-rich areas by particle analysis function.

Figure 4d: It would appear something is wrong with the lines given in the figure caption. That is, the lines given do not reproduce the lines shown in the graph, or something is left

unspecified (e.g., what is the delta on the x axis?). As currently written, I would say the ratio is higher for glass (1937.9 is higher than 1752.8), which is not what I think the authors are saying.

We apologize for the errors in the figure legend. Those were regression fittings from a different data set. The correct regression parameters for Fig. 4d should be: $F_{TFC} = 1.2083 \times F_{FM4-64} - 253.29$, $r = 0.6629$ for graphene; $F_{TFC} = 0.6273 \times F_{FM4-64} + 837.85$, $r = 0.5698$ for glass. We have made the correction in the figure legend.

Figure 6d: I am confused by this result. It appears to me that the authors have already presented results on synaptic vesicle density in Figure 4a (see my earlier comment where I argued that what the authors refer to as TFC puncta actually are FM4-64 puncta and hence synaptic vesicles). Those results showed a significant difference between glass and graphene, contrary to this results. It would be useful to the reader to understand the difference between these two sets of results.

Fig. 4a is meant to focus on TFC staining, not necessarily FM4-64 puncta or synaptic boutons. In Fig. 6d we specifically examine synaptic boutons identified by immunolabeling with a selective marker, Synaptophysin. Fig. 4d demonstrates that there were more patches of neuronal membrane with strong TFC labeling. This is consistent with an overall increase of neuronal surface cholesterol since the TFC patches were identified as the signal above a common fluorescence threshold. Fig. 6d shows that there was no significant difference in the linear density of synaptic boutons along neuronal processes between graphene and glass groups. We have expanded this in the main text.

p. 11 “confirmed single-Qdot labeling (Figure S5a) at this concentration”: I found this interpretation highly surprising. Surely there is a large number of doublets?

In this set of Qdot imaging experiments, one synaptic vesicle was loaded one Qdot, which we call “single-Qdot labeling” of individual synaptic vesicles⁶⁻⁹. Fig. S5a shows quantal analysis with Gaussian distributions representing single, double, and triple Qdots, which corresponds to one, two or three single Qdot-labeled synaptic vesicles in individual synaptic boutons defined by FM4-64. There are about 30% of synapses which contain more than one single Qdot-labeled synaptic vesicle. Nevertheless, we relied on the distinct changes of Qdot photoluminescence to identify individual fusion events, as demonstrated in Fig. S5b. The likelihood that two or more Qdot-labeled synaptic vesicles at the same synaptic boutons released at the same time with the same fusion mode is very small. We have clarified that in the text.

Figure 8b: As far as I understand, these results (partly) repeat those shown in Figure 3b. However, in Figure 3b the results for filippin staining on graphene and glass are largely similar (as already noted), while in Figure 8b they are substantially different. It would be helpful if the authors could elaborate on the difference.

As also explained earlier in our reply, Filipin intensities were measured at somas in Fig. 3b. In Fig. 8b, we measured Filipin intensities in neurites as we were focused on

presynaptic membrane cholesterol. We speculate that the larger magnitude of difference between the graphene and glass groups in Fig. 8b suggests that graphene mostly increases cholesterol in the plasma membrane - not intracellular membranes that are abundant in neuronal somas. We have revised the main text and methods section to point out the differences between Fig. 3b and Fig 8b.

Minor comments

p. 5: Please clarify that neuronal culture media contains cholesterol; it makes the experiment easier to understand.

We clarified that in the revised main text and the method section.

p. 6-7 “which is consistent with the formation of a protein corona on the graphene surface”: I understand that this was inserted to satisfy my comment, but I do not understand what it is supposed to mean.

We deposited Matrigel (a proteinaceous culture substrate) onto a bare graphene surface to mimic the formation of protein corona on bare graphene, and the Raman spectral difference between Matrigel-covered and the Matrigel-free areas indicates how a thick protein corona will change graphene’s Raman spectrum. We have revised the main text accordingly.

p. 7 first paragraph: Please rewrite for clarity vis-a-vis neurons, glial cells and hippocampal cultures.

We have rewritten that paragraph.

Figure 3 caption: What does the double asterisks mean? In the case of panel d, what is it that is statistically different? The mean GP?

It should be ** in the plot of Fig. 3b. We have corrected that.

Fig. 3d shows the distribution of GP values. We performed a Kolmogorov-Smirnov test on the two distributions, which were statistically different. We have added this to the figure legend for clarity.

Figure S3: Is the figure not mislabelled with respect to DAP5 and NBQX?

We apologize for the mistake. It is now corrected.

Figure 7a2, b2 and d2 and Figure S5d: It must be noted what red and black represent. I cannot find this in the captions.

They are now annotated in the figure legends.

Figure S4: I find the linear fits presented meaningless; clearly the data is not linear over the whole time duration. The authors should restrict the range to the latter part of the curves, where the data in fact is linear. Alternatively, simply remove the fits.

We have removed the fits for simplicity and clarity.

Figure S5a: The description is still not clear. Based on the authors' answer I understand what they mean, but the figure caption is not clear: "Quantal analysis [...] indicates that the mean photoluminescence intensity of loaded single Qdots was 378 ± 41 a.u.". In the graph the peak is around 3100. I would strongly urge the authors to go through the whole manuscript and sort out how they describe fluorescence intensities, background subtracted or not.

We are sorry about the confusion. We have revised the figure legend and related sentences in the main text.

p. 11: Reads "Figure 4d2"; should probably read "Figure 7d2".

Corrected.

p. 12: Reads "S8b&c"; should probably read "S5b&c".

Corrected.

Figure S5c: What is the grey line? I cannot find this in the caption.

That line was mistakenly included and is now removed. It was for a condition that we did not include in this manuscript.

p. 17 "Although graphene has been demonstrated to damage bacterial cell membranes it has been shown to have few adverse effects on eukaryotic cells [...]. This may be explained in part by [...] the fact that cholesterol is abundant in eukaryotic cell membranes but absent in most prokaryotic membranes.": I do not understand what the authors are suggesting. If there is no cholesterol graphene causes damage to cells?

Yes, it is our speculation that graphene causes damage to prokaryotic cells whose membranes do not have cholesterol. We have clarified that in the discussion.

p. 18 "this study": Meaning reference 21?

We have clarified this.

Reference

- 1 Dell, R. B., Holleran, S. & Ramakrishnan, R. Sample Size Determination. *Ilar Journal* **43**, 207-213 (2002).
- 2 Chu, B.-B. *et al.* Cholesterol Transport through Lysosome-Peroxisome Membrane Contacts. *Cell* **161**, 291-306, doi:<http://dx.doi.org/10.1016/j.cell.2015.02.019> (2015).
- 3 Veliev, F., Briancon-Marjollet, A., Bouchiat, V. & Delacour, C. Impact of crystalline quality on neuronal affinity of pristine graphene. *Biomaterials* **86**, 33-41, doi:10.1016/j.biomaterials.2016.01.042 (2016).
- 4 Rauti, R. *et al.* Graphene Oxide Nanosheets Reshape Synaptic Function in Cultured Brain Networks. *ACS Nano* **10**, 4459-4471, doi:10.1021/acsnano.6b00130 (2016).
- 5 Hu, W. *et al.* Protein corona-mediated mitigation of cytotoxicity of graphene oxide. *ACS Nano* **5**, 3693-3700, doi:10.1021/nn200021j (2011).
- 6 Park, H., Li, Y. & Tsien, R. W. Influence of synaptic vesicle position on release probability and exocytotic fusion mode. *Science* **335**, 1362-1366, doi:10.1126/science.1216937 (2012).
- 7 Zhang, Q., Cao, Y.-Q. & Tsien, R. W. Quantum dots provide an optical signal specific to full collapse fusion of synaptic vesicles. *Proceedings of the National Academy of Sciences* **104**, 17843-17848 (2007).
- 8 Zhang, Q., Harata, N. C. & Tsien, R. W. Using quantum dots to visualize the turnover of single synaptic vesicles. Program No. 365.13. 2005. *Abstract Viewer/Itinerary Planner*. Washington, DC: *Society for Neuroscience, Online*. (2005).
- 9 Gu, H., Lazarenko, R. M., Koktysh, D., Iacovitti, L. & Zhang, Q. A Stem Cell-Derived Platform for Studying Single Synaptic Vesicles in Dopaminergic Synapses. *Stem cells translational medicine* **4**, 887-893, doi:10.5966/sctm.2015-0005 (2015).

Reviewers' comments:

Reviewer #3 (Remarks to the Author):

While the authors have given several detailed answers to my queries, overall, I feel there are still too many unconvincing results and inconsistencies to justify publication at this time. The fact that it took two revisions to get to this point, with several questions still remaining, to me means the authors should reconsider their data in more detail, rather than making minor adjustments.

As background, perhaps I need to spell out my point regarding statistically significant vs "substantial" differences. I appreciate the authors' detailed reply, but I am afraid it misses my point. I was not suggesting that there was anything wrong with their statistical analysis. Indeed, as I noted, some of the results show impressively large sample sizes. Instead, what I was suggesting was that several results, while statistically significant, are not significant in absolute terms. A difference can be statistically significant without being important. Fig. S2f serves to show my point: Is there a difference between glass and graphene? Yes, of course, there is. Is it a major difference of broader implication? Maybe not. Cf. also Fig. 5b and d where the authors claim no differences.

If we go into more detail, one of the major claims of the manuscript is that growing neurons on graphene increases cholesterol. This is supposedly shown by Fig. 3a-b, but, as I have previously argued, the difference is small. Using the authors' values in their reply (129.8 and 139.5) shows that it is less than 8%. In this revision, the authors have an argument for why there is such a small difference (essentially Fig. 8b), but they still insist on showing Fig. 3b as their first evidence. Would it not be reasonable to make the case stronger, now that they have the data?

Additionally, should not Fig. 3b and 4b show, largely, the same results? They appear to be inconsistent. (Admittedly, it is not specified what the average in Fig. 4b is over, so it is not clear if they should be the same or not.)

Furthermore, while the authors have addressed the inconsistency between Fig. 3b and 8b, they have not addressed the one between Fig. 3c and 8c. As far as I understand their figure captions this cannot plausibly be explained by the same argument as applied to Fig. 3b and 8b. In general, I am also unsure of how to interpret any differences in these graphs. That is, what should I consider a small or a large difference? This is, of course, even more important when the results do not appear self-consistent (Fig. 3c and 8c).

I would also argue that the results in Figure 8b-d for neurons grown on glass "spiked" with cholesterol are not self-consistent. The membrane cholesterol is increased, say, 1.5-fold compared to graphene (Fig. 8b), while the response still does not reach the level of graphene in Figure 8d.

Continuing, Fig. 4 shows results based on "TFC-rich areas". Certainly the differences appear to be larger, but they are based upon an identification that I simply do not see a strong justification for. It is certainly explained better in the revised manuscript, but I simply do not see why those particular areas indicated by arrows are enriched with TFC, as opposed to the rest of the areas which exhibit TFC fluorescence. With the authors' reply it is a bit more clear, but a reader will not have access to that. This makes Fig. 4c equally difficult to evaluate.

With these many questions still remaining, for results central to the authors' main thesis, I really feel the authors need to make their case stronger.

Further comments:

Regarding corona formation: Again, the authors' reply misses my point. The authors are saying that the "cells can still access bare graphene without an intermediate protein corona". In order to

make that suggestion, they need the time scale of protein corona formation to be **slower** than the time scale of cell attachment. Figure 1c of Hu et al. clearly shows the opposite. The authors do not address this, but rather focus on a quote from Hu et al. Additionally, the quote is not even properly represented. Hu et al. say "GO-FBS protein binding reached equilibrium **within** 30 min" (my emphasis), while the authors say "Hu et al. recently showed that it took **about** 30 minutes for a stable protein corona to form" (my emphasis). Furthermore, in my last review I also noted that the time scale for 10 min was for complete saturation and the amount that remains after centrifugation. The authors do not address that. In essence, I think my expectation that the surface is covered by protein within minutes is more likely.

Figure 2e-f, Matrigel and corona: There is no suppression of the 2D and G peaks outside of cells, but presumably proteins will be adsorbed there. In other words, adsorption of these proteins will not be detected. In general, I feel the the authors' "analogy" between Matrigel and corona is strenuous, at best.

Minor comments:

Figure 3d: What is significantly different?

Figure 4a caption "Arrowheads indicate synaptic boutons": Please update.

Figure 4b caption "Average TFC staining intensity": Over what? Boutons? Cells? Fields of views?

Figure 4c caption "TFC puncta": Please update.

Figure 4d: What argument is this result meant to support?

Figure 6b caption "Arrowheads indicate Syp puncta": I see several other structures which I would consider to be puncta. It needs to be clarified whether the puncta identified by the authors are merely examples, or all of the structures they consider to be puncta.

Figure 6c: Surely they are different? Additionally, I do not understand how a two-tailed t test can be used to compare two distributions.

"Filipin staining confirmed a reduction of membrane cholesterol" This needs to be revised. As written it is not clear that it is in neurites.

"Filipin staining of neurites resulted in a larger difference between the graphene and glass groups than when measured at neuronal somas": This is not the appropriate place to address this.

Dear Dr. Bottari,

We sincerely thank the reviewer for a detailed and thorough review of our work. We have revised our manuscript taking into account the comments and questions of the reviewer, which we believe have improved the clarity and quality of our work. A detailed, point-by-point list of our response to the reviewer is given below. Manuscript changes are indicated directly in the manuscript in red typeface.

1. While the authors have given several detailed answers to my queries, overall, I feel there are still too many unconvincing results and inconsistencies to justify publication at this time. The fact that it took two revisions to get to this point, with several questions still remaining, to me means the authors should reconsider their data in more detail, rather than making minor adjustments.

We have carefully reconsidered our data in detail and made major efforts to reorganize our manuscript in order to ensure overall clarity and consistency. All of our data support the conclusion that graphene modifies cellular functions by increasing cellular cholesterol. Major/Important/Related changes are indicated in red typeface in the manuscript.

2. As background, perhaps I need to spell out my point regarding statistically significant vs “substantial” differences. I appreciate the authors’ detailed reply, but I am afraid it misses my point. I was not suggesting that there was anything wrong with their statistical analysis. Indeed, as I noted, some of the results show impressively large sample sizes. Instead, what I was suggesting was that several results, while statistically significant, are not significant in absolute terms. A difference can be statistically significant without being important. Fig. S2f serves to show my point: Is there a difference between glass and graphene? Yes, of course, there is. Is it a major difference of broader implication? Maybe not. Cf. also Fig. 5b and d where the authors claim no differences.

We agree that the differences in the original **Figure S2c&f** (currently **Figure S4c&f**) are not large enough to claim “significantly stronger in the graphene group at 3 and 7 DIV”. In the revised version, Cohen’s d , an effect size used to indicate the standardized difference between two means, was included where relevant. Both **Figure S4c** and **S4f** show small effect sizes (Cohen’s $d = 0.188$ and 0.146 for 3 and 7 DIV, respectively). In addition, the effect on sEPSC inter-event interval (**Figure 5c**) is much greater (Cohen’s $d = 1.3$), while both **Figure 5b** and **5d** have small effect sizes (Cohen’s $d = 0.10$ and 0.27 , respectively). We revised the related sentences. This does not affect our main claims.

3. If we go into more detail, one of the major claims of the manuscript is that growing neurons on graphene increases cholesterol. This is supposedly shown by Fig. 3a-b, but, as I have previously argued, the difference is small. Using the authors’ values in their reply (129.8 and 139.5) shows that it is less than 8%. In this revision, the authors have an argument for why there is such a small difference (essentially Fig. 8b), but they still insist on showing Fig. 3b as their first evidence. Would it not be reasonable to make the case stronger, now that they have the data?

We agree that the small difference in the previous **Figure 3b** is not strong enough to support our major claims. Somas contain a large number of membrane-bound organelles, whose cholesterol could also be labeled by Filipin, which could overshadow the cholesterol changes induced by graphene. Following the reviewer's suggestion, we replaced **Figure 3b** with the neurite Filipin staining data in the revised version, which is a ~ 27% increase on graphene and added related descriptions in the main text.

4. Additionally, should not Fig. 3b and 4b show, largely, the same results? They appear to be inconsistent. (Admittedly, it is not specified what the average in Fig. 4b is over, so it is not clear if they should be the same or not.)

Figure 4b (currently **Figure 4a2**) is averaged over all ROIs (representing neurite areas) in the same group. We used TFC in **Figure 4** to mimic cholesterol distribution in live cells, while Filipin staining in **Figure 3** is used to study endogenous cholesterol in fixed cells. Therefore, these two results are not directly comparable. We speculate that the reason for the relatively larger difference observed for TFC labeling may be that more TFC was attracted to the graphene surface very close to cells and served as a local supply to facilitate more cell membrane loading of TFC. Related discussion was added in the main text.

5. Furthermore, while the authors have addressed the inconsistency between Fig. 3b and 8b, they have not addressed the one between Fig. 3c and 8c. As far as I understand their figure captions this cannot plausibly be explained by the same argument as applied to Fig. 3b and 8b. In general, I am also unsure of how to interpret any differences in these graphs. That is, what should I consider a small or a large difference? This is, of course, even more important when the results do not appear self-consistent (Fig. 3c and 8c).

We apologize for the lack of explanation about GP value changes. **Figure 3c** are sample GP images and **Figure 3d** is the corresponding distribution plot of pixel GP values (GP values of every pixel in the analyzed images). In both **Figure 3d** and **8c**, the relative shifts of the peak positions between the glass and the graphene groups are similar, ~ 0.155.

Both the scale and the shape of the GP distributions are different since two different instrumentations were used for data acquisition in **Figure 3d** (a 405-nm laser source) vs. **8c** (an arc lamp with a Chroma D350x filter). In **Figure S8** (also shown below) we demonstrate the difference in the scale and the shape of GP distributions for the same FOV using the two different instrumentations.

Figure S8. GP values with two different excitation settings. (a & b) Sample images with 405 nm laser or Prior 200 light source and D350x filter from Chroma. Scale bar, 20 μm. (c) The distribution of GP pixel values in both conditions.

Figure S9. Cholesterol depletion by MβCD reduces GP values. (a) Sample images of the same field of view after treatment with 0, 0.5-mM 5-min, or 10-mM 30-min MβCD. Scale bar, 20 μm. (b) Distributions of GP values over individual image pixels ($n = 3$ FOVs, $N = 3$ batches for every treatment; for 0.5 and 10 mM in comparison to 0 mM, $p < 0.01$ and < 0.001 respectively, *Kolmogorov-Smirnov* test of the unbinned raw GP values).

To evaluate how GP differences are related to differences in neuronal membrane cholesterol, we performed GP imaging on neurons with weak or strong cholesterol depletion by MβCD (Zidovetzki and Levitan, 2007). As shown in **Figure S9**, the 0.5-mM MβCD (weak) shifted GP peak ~ 0.089 to the right and the subsequent 10-mM MβCD (strong) shifted GP peak ~ 0.102 further to the right. Therefore, we estimate that membrane cholesterol change by graphene or cholesterol manipulations were moderate (~ 0.155 is between 0.089 and 0.191).

This information was added to the main text and SOM.

6. I would also argue that the results in Figure 8b-d for neurons grown on glass “spiked” with cholesterol are not self-consistent. The membrane cholesterol is increased, say, 1.5-fold compared to graphene (Fig. 8b), while the response still does not reach the level of graphene in Figure 8d.

Cholesterol within the cell membrane is a regulatory factor but not the driving force or the determining factor for exocytotic release of neurotransmitter (i.e. “the response” measured by FM4-64 destaining in **Figure 8d**). SNARE protein complexes or other presynaptic proteins like Synaptotagmin are believed to control synaptic vesicle release. And the number of fusion sites available at the active zone of individual presynaptic terminals is also physically limited. Therefore, cholesterol’s facilitation of synaptic vesicle turnover may not be able to exceed the biological limit. It is also likely that there is a saturation point at which any further excess cholesterol cannot be functionally incorporated into the cell membrane and may be transported to the intracellular space. Therefore, this excess cholesterol may be measured via Filipin staining but not by GP imaging, which may explain our results in **Figure 8c** – moderate changes in GP value. C-laurdan only labels the plasma membrane, and only reports its fluidity. In addition, TFC is more water-soluble than cholesterol due to its BODIPY group, which may help to explain why TFC loading is much greater.

We added the related discussion in the main text.

Continuing, Fig. 4 shows results based on “TFC-rich areas”. Certainly the differences appear to be larger, but they are based upon an identification that I simply do not see a strong justification for. It is certainly explained better in the revised manuscript, but I simply do not see why those particular areas indicated by arrows are enriched with TFC, as opposed to the rest of the areas which exhibit TFC fluorescence. With the authors’ reply it is a bit more clear, but a reader will not have access to that. This makes Fig. 4c equally difficult to evaluate.

We understand the reviewer’s confusion. We now present the TFC staining data in **Figure 4** in a revised format to clarify our meaning and also include **Figure S2** (a schematic of ‘ROI’ selection for the TFC staining that was included in our last reply). This demonstrates that a threshold-based approach was used for the data presented in **Figure 4a2** and **4a3** without use of the ROIs that the arrowheads originally seem to present.

7. With these many questions still remaining, for results central to the authors’ main thesis, I really feel the authors need to make their case stronger.

As the reviewer suggested, we restructured our figures and descriptions to make our manuscript more convincing and our claims stronger.

8. Regarding corona formation: Again, the authors’ reply misses my point. The authors are saying that the “cells can still access bare graphene without an intermediate protein corona”. In order to make that suggestion, they need the time scale of protein corona formation to be

slower than the time scale of cell attachment. Figure 1c of Hu et al. clearly shows the opposite. The authors do not address this, but rather focus on a quote from Hu et al. Additionally, the quote is not even properly represented. Hu et al. say “GO–FBS protein binding reached equilibrium *within* 30 min” (my emphasis), while the authors say “Hu et al. recently showed that it took *about* 30 minutes for a stable protein corona to form” (my emphasis). Furthermore, in my last review I also noted that the time scale for 10 min was for complete saturation and the amount that remains after centrifugation. The authors do not address that. In essence, I think my expectation that the surface is covered by protein within minutes is more likely.

We apologize for our confusion. We agree with that there is likely protein deposition on the graphene surface. However, our results as well as recent findings of Veliev et. al. suggest that graphene still can exert cellular effects directly. It is quite difficult to fully exclude the existence of a protein corona from their configuration and thus their conclusions. The related information was added to the main text.

9. Figure 2e-f, Matrigel and corona: There is no suppression of the 2D and G peaks outside of cells, but presumably proteins will be adsorbed there. In other words, adsorption of these proteins will not be detected. In general, I feel the authors’ “analogy” between Matrigel and corona is strenuous, at best.

We agree with the reviewer that there can be some protein deposition on the graphene surface, which, unlike the thick gelled Matrigel coating, was not detectable by Raman. However, our results as well as recent findings of Veliev et. al. suggest that graphene still can exert cellular effects directly. We revised this argument in the main text.

Minor comments:

10. Figure 3d: What is significantly different?

The distribution of GP values from all pixels of all FOVs are different (*Kolmogorov-Smirnov* test, figure legend). We now clarify that.

11. Figure 4a caption “Arrowheads indicate synaptic boutons”: Please update.

Figure 4b caption “Average TFC staining intensity”: Over what? Boutons? Cells? Fields of views?

Figure 4c caption “TFC puncta”: Please update.

Figure 4d: What argument is this result meant to support?

The whole of **Figure 4** has been reformatted for clarity and the figure legend has been updated. The results in **Figure 4d** support our conclusion that graphene can increase cholesterol levels areas that already have locally high cholesterol concentrations. This is important because it may be possible that cholesterol could not be taken up in these membrane regions if it has already reached some physiological saturation point. It is also

meant to support our later data demonstrating that increased cholesterol results in synaptic potentiation on graphene.

12. Figure 6b caption “Arrowheads indicate Syp puncta”: I see several other structures which I would consider to be puncta. It needs to be clarified whether the puncta identified by the authors are merely examples, or all of the structures they consider to be puncta.

We have added notes in the figure legends in **Figure 6** to state that the arrowheads indicate example puncta.

13. Figure 6c: Surely they are different? Additionally, I do not understand how a two-tailed t test can be used to compare two distributions.

We apologize, the reviewer is correct that paired comparisons should not have been done without first performing an ANOVA analysis. We have now performed a Two-Way ANOVA with repeated measures followed by Bonferroni multiple comparisons tests. Indeed, the two groups are different ($p = 0.01$) for **Figure 6c**. However, none of the individual *post-hoc* comparisons are significantly different (similar to Yalgin et al., 2015).

14. “Filipin staining confirmed a reduction of membrane cholesterol” This needs to be revised. As written it is not clear that it is in neurites.

We revised the related sentence.

“Filipin staining confirmed a reduction of membrane cholesterol in neurites.”

15. “Filipin staining of neurites resulted in a larger difference between the graphene and glass groups than when measured at neuronal somas”: This is not the appropriate place to address this.

We deleted this statement.

We hope that this manuscript is now suitable for publication in *Nature Communications*.

Very best regards,

Qi Zhang

09/28/2017

Reference:

Yalgin, C., Ebrahimi, S., Delandre, C., Yoong, L.F., Akimoto, S., Tran, H., Amikura, R., Spokony, R., Torben-Nielsen, B., White, K.P., *et al.* (2015). Centrosomin represses dendrite branching by orienting microtubule nucleation. *Nat Neurosci* *18*, 1437-1445.

Zidovetzki, R., and Levitan, I. (2007). Use of cyclodextrins to manipulate plasma membrane cholesterol content: evidence, misconceptions and control strategies. *Biochim Biophys Acta* *1768*, 1311-1324.

Reviewers' comments:

Reviewer #3 (Remarks to the Author):

The revised manuscript has been much improved compared to previous versions and I now accept that there is evidence for the central claims of the manuscript. I am also pleased to read that we have now understood each other when it comes to the protein corona aspect and the distinction between "statistically significant difference" vs "important difference". Nevertheless, I still maintain that the presented evidence is not quite of the standard I would expect from a Nature research journal and the arguments in the manuscript still lacks clarity, as well as an attention to detail.

Some examples:

The non-cell experiments (Fig. 1), while perhaps not so important in the larger context of the manuscript, could still have needed some tightening. First, for the experiments reported in Fig. 1a why not simply suspend graphene together with cholesterol (or TFC) instead of medium? Second, I miss the control of filtered (PVP) medium without graphene. Also for connecting Fig. 1a and b, I miss the experiment reported in Fig. 1a performed with TFC.

Fig. 4: I feel the discussion of the results here should be nuanced. The authors are "spiking" the cells with the cholesterol analogue (TFC), not assessing the amount of cholesterol. It appears to me that this is two completely different things, and thus that statements such as "we then asked if graphene causes cholesterol increases within synaptic membranes" should be reviewed.

Also, while it is now much more clear what has been done, I feel that the quantification of the number of TFC-rich areas (Fig. 4a3, S2) is rather meaningless. This data is based on identifications such as those shown in Fig. S2 and I cannot tell in what sense the definition of such areas used by the authors is reasonable compared to any other definition.

Furthermore, in Fig. 4b3 I miss an argument as to why the TFC/FM4-64 ratio is the relevant measure, rather than simply the TFC fluorescence.

Moreover, please define what Delta FM4-64 is.

Finally, please clarify if c2 is without background subtraction and whether the numbers 8,787 and 11,050 are with or without background subtraction. I have raised this point before, but the lack of clarity keeps (re)appearing.

Fig. 5a: I wonder how representative the upper trace (in black) is, given the results aggregated in panel b.

Fig. 7: The quantification of the number of quantum dots is not great and I feel that should at least be acknowledged. If the background signal is 2719 a.u. and the fluorescence of a single quantum dot is 378 on the same scale, then this clearly leads to a concomitant problem when actually attempting to count quantum dots against the background.

Furthermore, in two panels of this figure cumulative distributions are presented. It would be useful to include an argument as to why this is the most relevant measure here.

Details:

Fig. 1a: please spell out what volume was used to resuspend the filtered graphene flakes. The assay apparently reports concentrations, so it needs to be specified that we can compare the different results on equal footing (I noted this already in my first review). In the materials and methods section would be enough.

Fig. 1b: Please spell out what "adjusted"/"corrected" fluorescence means. Was it a subtraction or division (or something else)? I notice that the axes that present "adjusted" fluorescences start at 100,000, rather than 0.

p. 7 paragraph starting line 142: Hippocampal cultures are not introduced. Furthermore, please rewrite the new sentence on "protein corona"; *I* understand what it is trying to say, but I do not think a reader who has not followed this review process will.

p. 7 line 159 "This result suggests that cells attach to graphene": This is an overly strong statement based on the experimental evidence. At most it says something happens.

Fig. 3/line 172: The results are variously referred to as from "neurites" or "dendrites". Please clarify what it is, and be consistent.

Line 176 "increased neuronal cholesterol": Should be "increased neuronal neurite cholesterol" (or "increased neuronal dendrite cholesterol", depending upon previous point).

Line 237-238 "not specific to any distance": On the contrary I would argue that it is specific to around 70 μm .

Line 240-241 "there were small increases in Syp staining intensity (Figure S4)": If the differences in Fig. 6c are described as "small" and "not specific to any distance", then I feel the differences in Fig. S4 are miniscule.

Line 251-252 "there was more cholesterol in the synaptic boutons of neurons growing on graphene (Figure 4b3)": The referenced figure presents a ratio of fluorescences, so either I am missing an argument here, or the statement needs revision.

p. 12: It is not clear where the referenced fractions (27, 30 and 26%) come from. They are nowhere specified in the figure nor caption (as far as I can tell) and thus need to be properly defined.

Line 296 "average numbers of total recycling vesicles": per synapse?

Line 896 "the unbinned raw GP values at individual pixels from each group": I do not understand what this means. The only meaning I can attach to it is that pixel number (0,0) of an image acquired for neurons grown on glass was compared with pixel number (0,0) of an image acquired for neurons grown on graphene, and so on for each pixel of the images. However, that would not seem very reasonable, so I am at a loss to what it actually means.

Fig. 6c: What do the errorbars represent?

Fig. 8: I am not sure I agree that the GP values are "similar between graphene and TFC-treated glass samples". (There was by the way a typo in my previous review: the comparison between Fig. 3c and 8c should have been between Fig. 3d and Fig. 8c.)

Fig. S3: The signals of the "5)" trace is described as "like those in the beginning", but they look very different compared to the "1)" trace. Some help in understanding this would be helpful.

Minor comments:

p. 3: "cellular activities or to delivery optical": Bad grammar.

p. 7: "our in vitro assays": All experiments reported in the manuscript (including the cell

experiments) are in vitro, so I think this should be rephrased for clarity.

p. 10: "14-18 DIV (Figure 6c and Supplementary Table S1)": Table S1 is for DIV 12, so this should be rephrased.

p. 12, second paragraph: The connection between the figures and the text is strenuous at best. That is, it is very difficult to follow the text while looking at the figure, or vice versa. Please consider either or both for clarity.

p. 13, line 299-302: I feel the correct position for these two sentences is in the previous paragraph.

Line 327 "Figure 8c-e": Should read "Figure 8d-e", I believe.

Line 323 "The avoid the cross-excitation...": This should come earlier in the paragraph.

Dear Dr. Bottari,

We sincerely thank the reviewer for a detailed and thorough review of our work. We are encouraged by comments of the reviewer. We have revised our manuscript taking into account the comments and questions of the reviewer, which we believe have improved the clarity and quality of our work. A detailed, point-by-point list of our response to the reviewer is given below. Manuscript changes are indicated directly in the manuscript in red typeface.

1. The non-cell experiments (Fig. 1), while perhaps not so important in the larger context of the manuscript, could still have needed some tightening. First, for the experiments reported in Fig. 1a why not simply suspend graphene together with cholesterol (or TFC) instead of medium? Second, I miss the control of filtered (PVP) medium without graphene. Also for connecting Fig. 1a and b, I miss the experiment reported in Fig. 1a performed with TFC.

Our culture media acts as the source of cholesterol in our system. Using media instead of exogenous cholesterol or TFC allowed us to directly examine if graphene could extract cholesterol from a system resembling our neuronal culture.

Graphene nanoflakes are hydrophobic; therefore, PVP (2 wt%) has been used to make them soluble in our culture media or Tyrode's solutions. In our experiments, we used media containing the same concentration of PVP as a control. As the low molecular weight of PVP means that it is not be able to be fractionated by the same columns we used for graphene, this control was not matched in terms of the filtered vs. unfiltered conditions for graphene.

The cholesterol assay we used is based on an enzyme-coupled reaction, in which cholesterol is oxidized by cholesterol oxidase to yield H_2O_2 . H_2O_2 was detected fluorescently using Amplex® Red reagent. TFC is not readily oxidized by cholesterol oxidase due to its BODIPY group. Additionally, its fluorescence would interfere with the assay. So we could not perform the same assay for TFC.

2. Fig. 4: I feel the discussion of the results here should be nuanced. The authors are “spiking” the cells with the cholesterol analogue (TFC), not assessing the amount of cholesterol. It appears to me that this is two completely different things, and thus that statements such as “we then asked if graphene causes cholesterol increases within synaptic membranes” should be reviewed.

We now follow the reviewer's recommendation and revise our statement in the main text.

“Given that TFC's distribution in surface and intracellular membranes mimics that of endogenous cholesterol (Hölttä-Vuori et al., 2008), we used TFC to examine graphene's effect uptake of a cholesterol surrogate. We labelled neurons with TFC, which distributes throughout the neuronal membrane (Figure 4a1). Intriguingly, neurons on graphene showed significantly increased TFC labeling (Figure 4a2).”

Also, while it is now much more clear what has been done, I feel that the quantification of the number of TFC-rich areas (Fig. 4a3, S2) is rather meaningless. This data is based on

identifications such as those shown in Fig. S2 and I cannot tell in what sense the definition of such areas used by the authors is reasonable compared to any other definition.

We now follow the reviewer's recommendation and remove both Fig. 4a3 and S2 plus related information in the main text and supplementary materials.

3. Furthermore, in Fig. 4b3 I miss an argument as to why the TFC/FM4-64 ratio is the relevant measure, rather than simply the TFC fluorescence.

Since TFC can insert in to all lipid membranes (including optically unresolvable synaptic vesicles), we presented the TFC data in this way to demonstrate that the overall increase in TFC loading was not due to an increase of synaptic vesicles (as measured by FM4-64). This would be equivalent to presenting the TFC fluorescence as the reviewer mentions, but also normalized by FM4-64 to account for vesicle number increases in the graphene group.

“As synaptic boutons generally have a higher cholesterol concentration than other parts of the neuronal membrane (Takamori et al., 2006; Wilhelm et al., 2014) which might thus limit the ability of the membrane to incorporate cholesterol analogue, we then asked if graphene could also increase TFC uptake within synaptic boutons. To this end, we measured TFC intensity in areas defined by FM4-64 labeling (i.e. synaptic boutons) (Figure 4b1&2, arrow heads), a far-red fluorescent dye which preferentially labels synaptic vesicles (Rouze and Schwartz, 1998; Vida and Emr, 1995). Since an increase of synaptic vesicles can increase total presynaptic membrane area and consequently TFC staining, we examined the relationship between TFC staining and FM4-64 staining (Figure 4b3, scatter plots and regression fits). The overall increase of TFC intensity relative to increases in FM4-64 intensity was higher on graphene (Figure 4b3), suggesting that graphene increases membrane cholesterol uptake regardless possible changes in synaptic vesicle numbers at synaptic boutons.”

Moreover, please define what Delta FM4-64 is.

Delta FM4-64 is the difference of FM4-64 fluorescence intensity before and after exhaustive stimulation, which represents the amount of FM4-64 loaded into releasable vesicles. This is now included in the text.

“ Δ FM 4-64 fluorescence intensity (see methods) vs. TFC intensity in FM4-64 defined synaptic boutons (both $n = 9$ FOVs, $N = 3$ batches for each group; $p < 0.05$, two-tailed t -test).”

“ Δ FM4-64 is defined here as the difference of FM4-64 fluorescence intensity before and after 4 rounds of exhaustive stimulation (Gaffield and Betz, 2006; Hoopmann et al., 2012).”

4. Finally, please clarify if c2 is without background subtraction and whether the numbers 8,787 and 11,050 are with or without background subtraction. I have raised this point before, but the lack of clarity keeps (re)appearing.

Both are background subtracted. We now clarify this in the figure legend.

“The average total Qdot photoluminescence intensities per synapse are $8,787 \pm 156$ a.u. for neurons on glass and $11,050 \pm 224$ a.u. for neurons on graphene ($n_{\text{glass}} = 187$ ROIs, $n_{\text{graphene}} = 211$

ROIs, $N = 4$ batches for each group; $p < 0.001$, Kolmogorov-Smirnov test) after background subtraction.”

Fig. 5a: I wonder how representative the upper trace (in black) is, given the results aggregated in panel b.

We have replaced Fig. 5a with a more representative trace.

Fig. 7: The quantification of the number of quantum dots is not great and I feel that should at least be acknowledged. If the background signal is 2719 a.u. and the fluorescence of a single quantum dot is 378 on the same scale, then this clearly leads to a concomitant problem when actually attempting to count quantum dots against the background.

We have acknowledged the issue in a supplemental discussion. Yes, the signal of a single Qdot is small relative to the background. This was necessary to ensure that our dynamic range was large enough to cover the signal for the total loading of all Qdots as there are 10s to 100s of recyclable vesicles inside of a diffraction limited area (i.e. synaptic boutons) – and thus 10s to 100s of Qdots. Thus, although the unitary brightness may be low, the signal of interest (recycling vesicles) can be measured with much greater certainty.

“Based on the unitary photoluminescence of a single Qdot (see supplementary discussion), we estimated that the average numbers of total recycling vesicles per synapse were 23.2 ± 0.4 on glass and 29.2 ± 0.6 on graphene, an ~26% increase (Figure 7c).”

“For the loading of the total releasable pool of synaptic vesicles, imaging conditions were optimized such that the dynamic range of our data would be covered but not saturated under the loading of Qdots to all recycling synaptic vesicles. Thus, the unitary brightness of a single Qdot is relatively low. This setting was optimized on a test coverslip and kept the same for both experimental conditions. Although the brightness of a single Qdot was low above the background, in this set of experiments many Qdots were loaded per synapse, which would thus allow greater certainty in the estimation of the number of total recycling vesicles per synapse. Furthermore, the estimate of the fluorescence of a single Qdot, even with a signal 378 a.u. above background, is easier to quantify given the ‘blinking’ property Qdots possess: all single quantifications were measured as the difference between a Qdot in its ‘off’ state and a Qdot in its ‘on’ state. A more detailed explanation of the use of blinking to help characterize single Qdot photoluminescence can be found in the reference (Zhang et al., 2009).”

5. Furthermore, in two panels of this figure cumulative distributions are presented. It would be useful to include an argument as to why this is the most relevant measure here.

We appreciate the suggestion and the argument is now included in the data analysis section of the methods. Cumulative distributions are commonly used to provide an overview of the intensity distribution from many synapses (see e.g. Sun and Turrigiano, 2011, Plotegher et al., 2017, and Thiagarajan et al., 2005).

“Cumulative distribution functions (FM dye and Qdot imaging) or histograms (GP imaging) were used for 2-group comparison of pooled values. Cumulative distributions are an accepted measure to provide an overview of intensity distributions from synapses (Plotegher et al., 2017; Sun and

Turrigiano, 2011; Thiagarajan et al., 2005) and more clearly demonstrate the overall trend of the individual data points than an average measure.”

Details:

6. Fig. 1a: please spell out what volume was used to resuspend the filtered graphene flakes. The assay apparently reports concentrations, so it needs to be specified that we can compare the different results on equal footing (I noted this already in my first review). In the materials and methods section would be enough.

We used 1 mL for resuspension. This information is now added in the methods section.

“1 mL aliquots of media were incubated as prepared or with the addition of 0.002 wt% PVP in PBS, or 260 ng/mL GNFs with 0.002 wt% PVP in PBS at 37°C and 5% CO₂ for 24 h. Graphene fractions were separated from media using Amicon centrifugal filters (100 kDa, Millipore) and resuspended in 1 mL of fresh media.”

Fig. 1b: Please spell put what “adjusted”/“corrected” fluorescence means. Was it a subtraction or division (or something else)? I notice that the axes that present “adjusted” fluorescences start at 100,000, rather than 0.

This refers to a division by the absorbance of graphene. The fluorescence was adjusted/corrected for the broad concentration-dependent absorbance by graphene (Fig. S1a). This information is now added in the methods section.

“averaged intensities were corrected for the broadband absorbance of either graphene or PVP across the emission spectra, as defined by: $\frac{F_{TFC\lambda}}{A_{graphene/PVP\lambda}}$, where F is the fluorescence emission value at a wavelength for TFC and A is the absorbance value at that same wavelength for graphene or PVP.”

The y-axis in Fig. 1b has been rescaled to start from zero.

7. p. 7 paragraph starting line 142: Hippocampal cultures are not introduced. Furthermore, please rewrite the new sentenced on “protein corona”; *I* understand what it is trying to say, but I do not think a reader who has not followed this review process will.

We have reorganized this section so that our culture method is introduced and explained here. We now follow the reviewer’s recommendation and rewrite our statement in the main text.

“It is well-established that nanomaterial surfaces will be covered by a variety of biomolecules (i.e. a protein corona) after introduction to any biological system. Although we omitted a much thicker artificial protein-coating layer (Matrigel) in plating our cultures, we cannot exclude the likelihood of the formation of a protein corona on the graphene surface.”

8. p. 7 line 159 “This result suggests that cells attach to graphene”: This is an overly strong statement based on the experimental evidence. At most it says something happens.

We now follow the reviewer’s recommendation and revise our statement in the main text.

“..., suggesting that plated neurons modulate graphene’s Raman spectra in the same manner that Matrigel does.”

9. Fig. 3/line 172: The results are variously referred to as from “neurites” or “dendrites”. Please clarify what it is, and be consistent.

We have clarified our nomenclature, and now use neurites throughout the text.

10. Line 176 “increased neuronal cholesterol”: Should be “increased neuronal neurite cholesterol” (or “increased neuronal dendrite cholesterol”, depending upon previous point).

We now follow the reviewer’s recommendation and revise our statement in the main text.

“increased neuronal neurite cholesterol”

11. Line 237-238 “not specific to any distance”: On the contrary I would argue that it is specific to around 70 μm .

We now follow the reviewer’s recommendation and revise our statement in the main text.

“the effect was small (see figure legend) and furthermore not significant at any distance. However, there was a non-significant increase on graphene at 68 μm .”

12. Line 240-241 “there were small increases in Syp staining intensity (Figure S4)”: If the differences in Fig. 6c are described as “small” and “not specific to any distance”, then I feel the differences in Fig. S4 are miniscule.

We now follow the reviewer’s recommendation and revise our statement in the main text.

“We found no differences in overall complexity (i.e. number of intersections); there were also only very small increases in Syp staining intensity (Figure S4).”

13. Line 251-252 “there was more cholesterol in the synaptic boutons of neurons growing on graphene (Figure 4b3)”: The referenced figure presents a ratio of fluorescences, so either I am missing an argument here, or the statement needs revision.

A more detailed description re: Fig. 4b3 can be found in our response to comment#3. After revisions to the text as a whole, we have removed this statement.

14. p. 12: It is not clear where the referenced fractions (27, 30 and 26%) come from. They are nowhere specified in the figure nor caption (as far as I can tell) and thus need to be properly defined.

The 30% decrease was from the sample inset in Fig. 7b2. We have now specified these values in the figure caption.

“Inset. Average FM1-43 fluorescence before destaining in the presence of high K^+ (pre-stimulation). $n_{\text{glass}} = 207$ ROIs, $n_{\text{graphene}} = 139$ ROIs, $N = 3$ batches for each group; $p < 0.01$, 2-sided t -test. The percentage reported in the main text is calculated from the inset values.”

“FM1-43 fluorescence during K^+ stimulation and (inset) average fluorescence values from the 170-180 s time window at the end of each decay curve ((black, glass; red, graphene. $n_{\text{glass}} = 207$ ROIs, $n_{\text{graphene}} = 139$ ROIs, $N = 3$ batches for each group; *** $p < 0.001$, two-tailed t -test). The percentage reported in the main text is calculated from the inset values.”

“FRF ratio (number of FRF events divided by number of total fusion events) during 1-min 10-Hz field stimulation (black, glass; red, graphene. $n_{\text{glass}} = 174$ ROIs, $n_{\text{graphene}} = 181$ ROIs, $N = 3$ batches for each group; *** $p < 0.001$, two-tailed t -test on the average FRF values from a 5-frame window at the end of each time course). The percentage (26%) reported in the main text is based on the average FRF values calculated for the statistical analysis.”

15. Line 296 “average numbers of total recycling vesicles”: per synapse?

Yes, we have clarified this where relevant.

“Based on the unitary photoluminescence of a single Qdot, we estimated that the average numbers of total recycling vesicles per synapse were 23.2 ± 0.4 on glass and 29.2 ± 0.6 on graphene”

16. Line 896 “the unbinned raw GP values at individual pixels from each group”: I do not understand what this means. The only meaning I can attach to it is that pixel number (0,0) of an image acquired for neurons grown on glass was compared with pixel number (0,0) of an image acquired for neurons grown on graphene, and so on for each pixel of the images. However, that would not seem very reasonable, so I am at a loss to what it actually means.

In GP images, we calculated a GP value at every pixel. These GP values were pooled to generate distributions for each treatment, which were used to test if the distribution of GP values between graphene and glass was different. We have expanded this explanation in the methods section.

“Kolmogorov-Smirnov test of the distributions of GP values, see data analysis section of the methods”

“For GP imaging data, all individual pixels from each field of view were pooled from each treatment condition as the GP distributions for the generation of test statistics.”

17. Fig. 6c: What do the errorbars represent?

All error bars are presented as mean \pm S.E.M. In Fig. 6c, this is the standard error mean of the pooled number of intersections at each distance from each treatment group. This has been clarified in the figure legend.

18. Fig. 8: I am not sure I agree that the GP values are “similar between graphene and TFC-treated glass samples”. (There was by the way a typo in my previous review: the comparison between Fig. 3c and 8c should have been between Fig. 3d and Fig. 8c.)

Our statement was based on a *Kolmogorov-Smirnov* test statistic ($p = 0.073$), although we acknowledge that there do appear to be differences in the distribution shape. This is now included in the figure caption.

“Distributions of GP values over individual image pixels ($n = 6$ FOVs, $N = 3$ batches for every group; for graphene vs. graphene + M β CD and glass vs. glass + TFC, both $p < 0.05$, for graphene vs. glass +TFC, $p = 0.073$, *Kolmogorov-Smirnov* test of the distributions of GP values, see data analysis section of the methods).”

Fig. S3: The signals of the “5)” trace is described as “like those in the beginning”, but they looks very different compared to the “1)” trace. Some help in understanding this would be helpful.

This is now more thoroughly explained in the caption for Fig. S3. Briefly, the difference is due to a difference in concentrations of Mg²⁺ in the external solution between (1) and (5). We have revised this section for clarity.

“Adding both D-AP5 and NBQX completely eliminates sEPSCs, and washout with 2 mM Mg²⁺ Tyrode’s solution recovers sEPSCs that are largely mediated by fast AMPAR components which can also be seen in the traces in (1). However, 0 mM Mg²⁺, as shown in (1), acts to remove the blockade of NMDA receptors that exists in normal physiological solution (Mangan and Kapur, 2004). Thus, the contribution of NMDA-receptor mediated currents is more readily visible in (1) than in (5).”

Minor comments:

19. p. 3: “cellular activities or to delivery optical”: Bad grammar.

Corrected.

“its superior carrier mobility enables graphene-based electrodes to detect electrochemical changes associated with a variety of cellular activities or to deliver optical or electrical stimuli.”

p. 7: “our in vitro assays”: All experiments reported in the manuscript (including the cell experiments) are in vitro, so I think this should be rephrased for clarity.

Rephrased.

“However, our cell-free assays demonstrated that cholesterol enrichment on graphene still occurs even in the likely presence of a protein corona after 24-hour incubation (Figure 1a and Figure S1).”

p. 10: “14-18 DIV (Figure 6c and Supplementary Table S1)”: Table S1 is for DIV 12, so this should be rephrased.

Rephrased.

“Although there was an overall increase in dendritic complexity on graphene between 12-18 DIV (Figure 6c and Supplementary Table S1)”

20. p. 12, second paragraph: The connection between the figures and the text is strenuous at best. That is, it is very difficult to follow the text while looking at the figure, or vice versa. Please consider either or both for clarity.

We have now modified that paragraph by rearranging this section to improve the clarity and readability with respect to the figure, and also by moving some of the methods description to the methods section in the text. We also more directly refer to the loading figure.

“To further elucidate the mechanisms underlying this presynaptic potentiation, we performed single vesicle imaging using quantum dots (Qdots), an approach which provides a more precise estimate of releasable synaptic vesicle amounts and their release probability. We began by loading Qdots into all releasable vesicles (the total releasable pool, TRP)^{60,61} using a combination of a high concentration of Qdots (100 nM) and strong stimulation (2-minute 90 mM-K⁺) (**Figure 7c1**). Based on the unitary photoluminescence of a single Qdot (see supplementary discussion), we estimated that the **average numbers of total releasable pool vesicles per synapse** were 23.2 ± 0.4 on glass and 29.2 ± 0.6 on graphene, a ~26% increase (**Figure 7c2**).”

21. p. 13, line 299-302: I feel the correct position for these two sentences is in the previous paragraph.

We have moved that sentence to the previous paragraph.

22. Line 327 “Figure 8c-e”: Should read “Figure 8d-e”, I believe.

Corrected.

23. Line 323 “The avoid the cross-excitation...”: This should come earlier in the paragraph.

This was moved to earlier in the paragraph.

We hope that this manuscript is now suitable for publication in *Nature Communications*.

Very best regards,

Qi Zhang

References

- Gaffield, M.A., and Betz, W.J. (2006). Imaging synaptic vesicle exocytosis and endocytosis with FM dyes. *Nat Protoc* 1, 2916-2921.
- Höltkä-Vuori, M., Uronen, R.-L., Repakova, J., Salonen, E., Vattulainen, I., Panula, P., Li, Z., Bittman, R., and Ikonen, E. (2008). BODIPY-Cholesterol: A New Tool to Visualize Sterol Trafficking in Living Cells and Organisms. *Traffic* 9, 1839-1849.
- Hoopmann, P., Rizzoli, S.O., and Betz, W.J. (2012). Imaging synaptic vesicle recycling by staining and destaining vesicles with FM dyes. *Cold Spring Harb Protoc* 2012, 77-83.
- Mangan, P.S., and Kapur, J. (2004). Factors Underlying Bursting Behavior in a Network of Cultured Hippocampal Neurons Exposed to Zero Magnesium. *Journal of Neurophysiology* 91, 946.
- Plotegher, N., Berti, G., Ferrari, E., Tessari, I., Zanetti, M., Lunelli, L., Greggio, E., Bisaglia, M., Veronesi, M., Giroto, S., *et al.* (2017). DOPAL derived alpha-synuclein oligomers impair synaptic vesicles physiological function. *Sci Rep* 7, 40699.
- Rouze, N.C., and Schwartz, E.A. (1998). Continuous and Transient Vesicle Cycling at a Ribbon Synapse. *The Journal of Neuroscience* 18, 8614.
- Sun, Q., and Turrigiano, G.G. (2011). PSD-95 and PSD-93 play critical but distinct roles in synaptic scaling up and down. *J Neurosci* 31, 6800-6808.
- Takamori, S., Holt, M., Stenius, K., Lemke, E.A., Grønborg, M., Riedel, D., Urlaub, H., Schenck, S., Brügger, B., Ringler, P., *et al.* (2006). Molecular Anatomy of a Trafficking Organelle. *Cell* 127, 831-846.
- Thiagarajan, T.C., Lindskog, M., and Tsien, R.W. (2005). Adaptation to synaptic inactivity in hippocampal neurons. *Neuron* 47, 725-737.
- Vida, T.A., and Emr, S.D. (1995). A new vital stain for visualizing vacuolar membrane dynamics and endocytosis in yeast. *The Journal of Cell Biology* 128, 779.
- Wilhelm, B.G., Mandad, S., Truckenbrodt, S., Krohnert, K., Schafer, C., Rammner, B., Koo, S.J., Classen, G.A., Krauss, M., Haucke, V., *et al.* (2014). Composition of isolated synaptic boutons reveals the amounts of vesicle trafficking proteins. *Science* 344, 1023-1028.
- Zhang, Q., Li, Y., and Tsien, R.W. (2009). The Dynamic Control of Kiss-And-Run and Vesicular Reuse Probed with Single Nanoparticles. *Science* 323, 1448-1453.

Reviewers' Comments:

Reviewer #1:

Remarks to the Author:

This paper is significantly improved from previous versions of the manuscript and the comments from previous reviews have been thoroughly addressed. However this reviewer has identified recent papers that call into question the use of TopFluor Cholesterol (TFC) as a surrogate for cholesterol trafficking. Therefore, since the experiments carried out by the authors have used TFC to study the effects of graphene on cholesterol the authors should provide additional data to support the use of TFC in their experiments. The findings of this paper are interesting but due to the lessened impact of the results because of the use of a cholesterol analog, the excitement of this reviewer is lessened significantly.

1. In Fig. 1b, have the authors evaluated BODIPY itself for adsorption to GFs as a control? The hydrophobic nature of BODIPY may lead to a similar result. Fluorescently-labeled cholesterol analogs have a similarly sized dye molecule attached where the behavior of the analog could significantly differ from that of unmodified cholesterol.

2. Fig. 4. Although the authors have added an additional description regarding the use of TFC instead of cholesterol only, there has been recent discussion in the literature that questions that use of BODIPY-Cholesterol. The authors should address the findings of these papers to provide further rationale for why TFC is acceptable compared to cholesterol itself or other cholesterol analogs. Examples are:

a. (Sezgin et al. A comparative study on fluorescent cholesterol analogs as versatile cellular reporters. *Journal of Lipid Research*. 2016). "TF-Chol, an analog that had so far been regarded to perform well in cellular assays, did not behave appropriately in the intracellular trafficking assay and did not show good performance in STED-based experiments, yet indicated good properties for testing different membrane phases."

b. Another more recent paper by Hölttä-Vuori (Use of BODIPY-Cholesterol (TF-Chol) for Visualizing Lysosomal Cholesterol Accumulation. *Traffic*. 2016) also describes the lack of suitability of using TF-Chol to measure cholesterol deposition. The description of this result is concerning given the same authors seem to be retracting their previous support for the use of TF-Chol as a cholesterol analog. This is not in agreement with the statement by the authors "Given that TFC's distribution in surface and intracellular membranes mimics that of endogenous cholesterol (Hölttä-Vuori et al., 2008), we used TFC to examine graphene's effect uptake of a cholesterol surrogate. We labelled neurons with TFC, which distributes throughout the neuronal membrane (Figure 4a1). Intriguingly, neurons on graphene showed significantly increased TFC labeling (Figure 4a2)."

3. Leading comments in the discussion such as "for the first time" should be removed.

Reviewers' comments: Reviewer #1 (Remarks to the Author):

This paper is significantly improved from previous versions of the manuscript and the comments from previous reviews have been thoroughly addressed. However this reviewer has identified recent papers that call into question the use of TopFluor Cholesterol (TFC) as a surrogate for cholesterol trafficking. Therefore, since the experiments carried out by the authors have used TFC to study the effects of graphene on cholesterol the authors should provide additional data to support the use of TFC in their experiments. The findings of this paper are interesting but due to the lessened impact of the results because of the use of a cholesterol analog, the excitement of this reviewer is lessened significantly.

1. In Fig. 1b, have the authors evaluated BODIPY itself for adsorption to GFs as a control? The hydrophobic nature of BODIPY may lead to a similar result. Fluorescently-labeled cholesterol analogs have a similarly sized dye molecule attached where the behavior of the analog could significantly differ from that of unmodified cholesterol.

As the reviewer suggested, we now compare BODIPY and TFC using the same conditions as originally shown in Fig. 1b. We observe very little and far less substantial quenching of BODIPY than TFC in the presence of GFs. Fig.1b and the main text are now updated to include the BODIPY dataset.

Redacted

Additionally, Fig. 1a in the main text serves as further support that GFs extract cholesterol from the culture media.

Our data collectively suggest that GFs interact with both endogenous cholesterol and TFC; and the interaction with TFC is most likely attributed to TFC's cholesterol group.

2. Fig. 4. Although the authors have added an additional description regarding the use of TFC instead of cholesterol only, there has been recent discussion in the literature that questions that use of BODIPY-Cholesterol. The authors should address the findings of these papers to provide further rationale for why TFC is acceptable compared to cholesterol itself or other cholesterol analogs. Examples are:

a. (Sezgin et al. A comparative study on fluorescent cholesterol analogs as versatile cellular reporters. Journal of Lipid Research. 2016). "TF-Chol, an analog that had so far been regarded to perform well in cellular assays, did not behave appropriately in the

intracellular trafficking assay and did not show good performance in STED-based experiments, yet indicated good properties for testing different membrane phases.”

b. Another more recent paper by Hölttä-Vuori (Use of BODIPY-Cholesterol (TF-Chol) for Visualizing Lysosomal Cholesterol Accumulation. Traffic. 2016) also describes the lack of suitability of using TF-Chol to measure cholesterol deposition. The description of this result is concerning given the same authors seem to be retracting their previous support for the use of TF-Chol as a cholesterol analog. This is not in agreement with the statement by the authors “Given that TFC’s distribution in surface and intracellular membranes mimics that of endogenous cholesterol (Hölttä-Vuori et al., 2008), we used TFC to examine graphene’s effect uptake of a cholesterol surrogate. We labelled neurons with TFC, which distributes throughout the neuronal membrane (Figure 4a1). Intriguingly, neurons on graphene showed significantly increased TFC labeling (Figure 4a2).”

We appreciate the reviewer’s mention of two recent publications regarding the use of TFC in studying cholesterol. Both studies note that TFC behaves differently from cholesterol in terms of intracellular trafficking, especially its deposition in lysosomes. Therefore, they conclude that TFC is not suitable for studying lysosomal storage diseases like Niemann-Pick Type C disease. However, both studies also acknowledge the utility of TFC in membrane-based experiments, especially its similarity to cholesterol in membrane partitioning. As we focused our study on the effect of graphene on membrane cholesterol, these two reports actually support our usage of TFC. We now cite these publications in our revised manuscript and have addressed our justification for the use of TFC in our revised text.

Both references used a higher concentration of TFC and longer loading times to study TFC’s intracellular trafficking and lysosomal deposition. In our case, we incubated neurons or 3T3 cells with 1 μM TFC (in the culture media) for 1 hour, which was sufficient to demonstrate membrane loading. Those two studies used 4-hour incubation with 5 $\mu\text{g}/\text{mL}$ ($\sim 8.7\mu\text{M}$) TFC. Hölttä-Vuori et al. further demonstrated that in order to obtain lysosomal deposition of TFC similar to endogenous cholesterol it was better to incubate cells in lipoprotein-deprived media for 18 hours (or longer). Therefore, our TFC loading condition was more suitable for studying membrane cholesterol.

We do not use STED in our study, so the fact that TFC did not perform well in STED was not a concern in our selection of a cholesterol analogue and did not affect our conclusions.

3. Leading comments in the discussion such as “for the first time” should be removed.

We have corrected this and all locations in the discussion with leading comments.

Reference:

Digman, M.A., Caiolfa, V.R., Zamai, M., and Gratton, E. (2008). The phasor approach to fluorescence lifetime imaging analysis. *Biophys J* 94, L14-16.

Reviewers' Comments:

Reviewer #1:

Remarks to the Author:

The authors have adequately addressed the raised comments and clarified their findings more thoroughly. It is now recommended to accepted this paper without further revisions.

P9. L192. Punctuation error - space after 'labelling .'